# Gradient Descent Is Optimal Under Lower Restricted Secant Inequality And Upper Error Bound

**Charles Guille-Escuret**
Mila, Université de Montréal
`guillech@mila.quebec`

**Adam Ibrahim**
Mila, Université de Montréal
`adam.ibrahim@mila.quebec`

**Baptiste Goujaud**
Ecole Polytechnique
`baptiste.goujaud@gmail.com`

**Ioannis Mitliagkas**
Mila, Université de Montréal,
Canada CIFAR AI chair
`ioannis@mila.quebec`

## Abstract

The study of first-order optimization is sensitive to the assumptions made on the objective functions. These assumptions induce complexity classes which play a key role in worst-case analysis, including the fundamental concept of algorithm optimality. Recent work argues that strong convexity and smoothness—popular assumptions in literature—lead to a pathological definition of the condition number [12]. Motivated by this result, we focus on the class of functions satisfying a lower restricted secant inequality and an upper error bound. On top of being robust to the aforementioned pathological behavior and including some non-convex functions, this pair of conditions displays interesting geometrical properties. In particular, the necessary and sufficient conditions to interpolate a set of points and their gradients within the class can be separated into simple conditions on each sampled gradient. This allows the performance estimation problem (PEP, [7, 33]) to be solved analytically, leading to a lower bound on the convergence rate that proves gradient descent to be exactly optimal on this class of functions among all first-order algorithms.

## 1 Introduction

The typical framework to study convergence properties of first-order algorithms in the context of machine learning is to first establish a class of objective functions to optimize through assumptions usually bound to a constant, such as $L$-smoothness and $\mu$-strong convexity. A tuning prescription of an algorithm is then made based on the constants, (e.g. step size $\alpha = \frac{2}{\mu+L}$ in the case of the gradient descent method on smooth and strongly convex functions), and finally a worst-case convergence rate can be derived for this algorithm when using this tuning prescription. In some cases, a lower bound on the achievable worst-case convergence rate can also be derived, leading to the theoretical optimality of an algorithm on the considered class of function, for instance the Nesterov accelerated gradient method [27] is known to be optimal up to a constant on strongly convex and smooth functions.

However in [12], the authors establish that such framework and its derived results are very sensitive to the choice of assumptions, and that strong convexity and smoothness can exhibit pathological behaviors leading to conservative tuning and arbitrarily sub-optimal convergence rates, even when the resulting algorithm achieves theoretical worst-case optimality on this class of functions. Furthermore, they propose a set of more robust alternative conditions. In this work, we focus on a specific pair of such alternative conditions : lower restricted secant inequality ($\mathrm{RSI}^-$) and upper error bounds ($\mathrm{EB}^+$). Our main contribution is to show that the gradient descent (GD) method with a certain tuning

36th Conference on Neural Information Processing Systems (NeurIPS 2022).

is exactly optimal on the classes of objective functions induced by these conditions, confirming that optimality results are highly sensitive to the choice of conditions. Another consequence is that no algorithm can accelerate on this class of functions, implying that additional assumptions are required to explain the practical efficiency of accelerated methods.

**Notation**  Let $\mathcal{F}$ the set of differentiable functions from $\mathbb{R}^d$ to $\mathbb{R}$ that admit a convex set of global minima $X_f^*$. We focus on the problem of optimizing a function $f \in \mathcal{F}$, i.e. finding $x \in X_f^*$. For any $x \in \mathbb{R}^d$, we denote $x_f^*$ the orthogonal projection of $x$ on $X_f^*$. By abuse of notation, when the context is not ambiguous, we will simply denote $x_f^*$ as $x^*$.

We call gradient descent (GD) the standard optimization algorithm based on the following update, where $\alpha$ is the step size :
$$x_{i+1} = x_i - \alpha \nabla f(x_i)$$

We call *first-order algorithm* all $\mathcal{A}$ that consider past iterates, function values and gradients and output a next iterate. Formally, $\mathcal{A}$ can be seen as a sequence of functions $\{\mathcal{A}_n \mid n \in \mathbb{N}\}$ such that for any $n \in \mathbb{N}$, $\mathcal{A}_n$ is a function defined on $\left(\mathbb{R}^d \times \mathbb{R} \times \mathbb{R}^d\right)^{n+1}$ with values in $\mathbb{R}^d$. Under that formalism, applying $\mathcal{A}$ to optimize an objective function $f$ starting in $x_0 \in \mathbb{R}^d$ generates a sequence of iterates $(x_i)_i$ such that $\forall i, x_{i+1} = \mathcal{A}_i\left((x_j, f(x_j), \nabla f(x_j))_{j \leq i}\right)$. Note that we do not require the iterates of $\mathcal{A}$ to lie within the span of observed gradients as it is often the case in the literature.

**Outline**  Section 2 introduces $\mathrm{RSI}^-$ and $\mathrm{EB}^+$ and provides some basic properties and motivation. Section 3 discusses related works in the literature. Section 4 defines and establishes the necessary and sufficient interpolation conditions for $\mathrm{RSI}^-$ and $\mathrm{EB}^+$, which is a key element of our analysis. Section 5 proves the lower bound on the convergence rate of first-order algorithms under $\mathrm{RSI}^-$ and $\mathrm{EB}^+$, and finally Section 6 concludes our work. Detailed proofs are provided in the appendix.

## 2  Lower restricted secant inequality and upper error bounds

We now define $\mathrm{RSI}^-$ and $\mathrm{EB}^+$ and discuss some basic properties.

**Definition 2.1 (Lower restricted secant inequality)**  *Let $f \in \mathcal{F}$ and $\mu > 0$*
$$f \in \mathrm{RSI}^-(\mu) \Leftrightarrow \forall x \in \mathbb{R}^d, \langle \nabla f(x) \mid x - x^* \rangle \geq \mu \|x - x^*\|_2^2$$

Intuitively, $\mathrm{RSI}^-(\mu)$ enforces that the further $x$ is from $X_f^*$, the stronger the gradient of $f$ in $x$ must be in the opposite direction of $X_f^*$.

**Remark 2.2**  $\mathrm{RSI}^-(\mu)$ *includes non-convex functions. However, it prevents flat landscapes outside of $X_f^*$, and requires $f$ to increase at least quadratically with the distance to $X_f^*$, as established in [12] :*
$$f \in \mathrm{RSI}^-(\mu) \Rightarrow \forall x \in \mathbb{R}^d, f(x) - f^* \geq \frac{\mu}{2} \|x - x^*\|_2^2$$

**Definition 2.3 (Upper Error Bounds)**  *Let $f \in \mathcal{F}$ and $L > 0$*
$$f \in \mathrm{EB}^+(L) \Leftrightarrow \forall x \in \mathbb{R}^d, \|\nabla f(x)\|_2 \leq L \|x - x^*\|_2$$

$\mathrm{EB}^+(L)$ thus enforces that the gradient of $f$ is controlled by the distance to $X_f^*$.

**Remark 2.4**  *$L$-smoothness implies $\mathrm{EB}^+(L)$ and $\mu$-strong convexity implies $\mathrm{RSI}^-(\mu)$. However, one must be careful before claiming that $\mathrm{RSI}^-$ and $\mathrm{EB}^+$ are respectively weaker than strong convexity and smoothness : for $\mu_0 > \mu_1$, $\mathrm{RSI}^-(\mu_0)$ is neither stronger or weaker than $\mu_1$-strong convexity, and for $L_0 < L_1$, $\mathrm{EB}^+(L_0)$ is neither stronger or weaker than $L_1$-smoothness.*

Therefore, even when the objective function is smooth and strongly convex, considering convergence results under $\mathrm{RSI}^-$ and $\mathrm{EB}^+$ is relevant, as we might obtain better constants $\mu$ and $L$. Many

convergence results depend of the condition number $\kappa = \frac{L}{\mu}$. Better constants leads to a better condition number, and thus a potentially better convergence rate (including when the dependence in the condition number $\kappa$ is the same or worse). As a consequence, machine learning problems with strongly convex and smooth objective functions are all potential applications of results under $\mathrm{RSI}^-$ and $\mathrm{EB}^+$, provided we can obtain better constants under these conditions.

---

**Prop 1 (convergence rate of GD under** $\mathrm{RSI}^-$ **and** $\mathrm{EB}^+$**)** *Let* $f \in \mathrm{RSI}^-(\mu) \cap \mathrm{EB}^+(L)$. *Then gradient descent with learning rate* $\alpha = \frac{\mu}{L^2}$ *on* $f$ *will guarantee the following convergence rate:*

$$\|x_i - x_i^*\|_2^2 \leq \left(1 - \frac{\mu^2}{L^2}\right)^i \|x_0 - x_0^*\|_2^2 \tag{1}$$

*Moreover, when the learning rate is set to* $\alpha_0 = \frac{1}{2\mu}$ *on the first step, and* $\alpha = \frac{\mu}{L^2}$ *on every other step, gradient descent guarantees the following convergence rate:*

$$\|x_i - x_i^*\|_2^2 \leq \frac{\|\nabla f(x_0)\|_2^2}{4\mu^2} \left(1 - \frac{\mu^2}{L^2}\right)^{i-1} \tag{2}$$

---

*Proof.* When $\alpha = \frac{\mu}{L^2}$, we have

$$
\begin{aligned}
\left\|x_{i+1} - x_{i+1}^*\right\|_2^2 &\leq \|x_{i+1} - x_i^*\|_2^2 \\
&= \|x_i - \alpha \nabla f(x_i) - x_i^*\|_2^2 \\
&= \|x_i - x_i^*\|_2^2 - 2\alpha \langle \nabla f(x_i) \mid x_i - x_i^* \rangle + \alpha^2 \|\nabla f(x_i)\|_2^2 \\
&\leq \left(1 - 2\alpha\mu + L^2\alpha^2\right) \|x_i - x_i^*\|_2^2 \\
&= \left(1 - \frac{\mu^2}{L^2}\right) \|x_i - x_i^*\|_2^2 ,
\end{aligned}
\tag{3}
$$

which proves (1). To prove (2), we simply note that when using $\alpha = \frac{1}{2\mu}$, we have:

$$
\begin{aligned}
\|x_1 - x_1^*\|_2^2 &\leq \|x_0 - x_0^*\|_2^2 - 2\alpha \langle \nabla f(x_0) \mid x_0 - x_0^* \rangle + \alpha^2 \|\nabla f(x_0)\|_2^2 \\
&\leq \frac{1}{\mu} \langle \nabla f(x_0) \mid x_0 - x_0^* \rangle - \frac{1}{\mu} \langle \nabla f(x_0) \mid x_0 - x_0^* \rangle + \frac{\|\nabla f(x_0)\|_2^2}{4\mu^2} \\
&= \frac{\|\nabla f(x_0)\|_2^2}{4\mu^2} .
\end{aligned}
\tag{4}
$$

∎

Interestingly, $\mathrm{RSI}^-$ and $\mathrm{EB}^+$ are direct bounds on the two additional terms obtained by developing $\|x_i - \alpha \nabla f(x_i) - x_i^*\|_2^2$, leading to an extremely simple proof. On the intuitive level, $\mathrm{RSI}^-$ lower bounds the gain from stepping in the direction of $X_f^*$, while $\mathrm{EB}^+$ upper bounds the error coming from the component of the gradient orthogonal to that direction.

**Remark 2.5** *The literature gives a worst case convergence rate of gradient descent on* $\mu$-*strongly convex and* $L$-*smooth functions of* $\|x_i - x_i^*\|_2^2 \leq \left(1 - \frac{2\mu}{\mu+L}\right)^{2i} \|x_0 - x_0^*\|_2^2$, *using the step size* $\alpha = \frac{2}{\mu+L}$ *[28, 30]. While this rate is better for a fixed* $\mu$ *and* $L$, *we again emphasize that the constants* $\mu$ *and* $L$ *may be very different depending on the chosen conditions, and thus these two rates can not directly be compared.*

**Motivation:** we empirically verify in Appendix A that the optimization path of a ResNet18 [16] trained for classification on CIFAR10 [20] verifies the interpolation conditions of $RSI^- \cap EB^+$, which are introduced in Section 4. This result guarantees the existence of a function in $RSI^- \cap EB^+$ which exactly interpolates the observed gradients, and thus all convergence guarantees of $RSI^- \cap EB^+$ naturally apply in this practical deep learning setting. $RSI^- \cap EB^+$ therefore provides linear convergence guarantees while being empirically applicable to neural networks (although with respects to a local minima), an impressive feat given the highly non-convex nature of neural networks loss functions.

## 3   Related Work

Throughout the literature, many choices of assumptions have been used to study first-order optimization. Most assumptions fall into one of two categories : *lower conditions* and *upper conditions* that respectively take the form of a lower and an upper bound on properties of the objective function. For instance, strong convexity lower bounds the curvature of the objective function and is thus a *lower condition*. Similarly, smoothness is an *upper condition*.

Lower conditions have been the most extensively studied assumptions, such as Polyak-Łojasiewicz [29], *local-quasi-convexity* [15], *weak quasi-convexity* [14], *quadratic growth* [2, 4, 18], *Kurdyka-Łojasiewicz* [21, 3], *optimal strong convexity* [22, 25, 8], *weak strong convexity* [19, 26], *error bounds* [24]. Some recent works have explored the relations between these lower conditions [19, 38]. In this work we focus on the *restricted secant inequality* $(\mathrm{RSI}^-)$ (which we denote as *lower restricted secant inequality* to differentiate it from its upper bound equivalent) which was introduced in [39], and has been used (along with its convex extension *restricted strong convexity*) in many recent theoretical derivations of linear convergence rates [36, 31, 37].

On the contrary, because most machine learning objective functions are naturally smooth, fewer works have explored alternatives to smoothness. However as discussed in Remark 2.4, it is still relevant to study these alternatives on smooth objectives due to potentially better conditioning. The most notable ones in the literature are *local smoothness* [15], *restricted smoothness* [1], *relative smoothness* [23, 13, 40], *weak-smoothness* [14], *expected smoothness* [11]. In [12], the authors argue that *lower conditions* can be naturally translated into equivalent *upper conditions* by changing the lower bound into an upper bound, and vice-versa. Subsequently, they introduce a set of *upper* equivalent to existing *lower conditions*, such as *upper error bounds* $\mathrm{EB}^+$, the natural *upper* equivalent to *error bounds* from [24]. Throughout this work, we focus on $\mathrm{EB}^+$ as an *upper condition*.

Finally, a key to our analysis are the necessary and sufficient interpolation conditions of $\mathrm{RSI}^-(\mu) \cap \mathrm{EB}^+(L)$ (Section 4). The search of these conditions has been largely motivated by Performance Estimation Problems (PEP), introduced in [33] and with many recent successful applications [17, 32, 6, 35, 34]. PEP is a framework for computer-assisted worst-case convergence analysis, that only requires necessary and sufficient interpolation conditions for the considered class of objective functions. In the case of $\mathrm{RSI}^- \cap \mathrm{EB}^+$ however, we found the interpolation conditions to be *independent* (see Section 4), which made the worst-case analysis directly solvable analytically.

## 4   Interpolation conditions

In this section we provide and discuss the necessary and sufficient interpolation conditions for $\mathrm{RSI}^- \cap \mathrm{EB}^+$. Their importance stems from the framework used in PEP and in this work to analyze worst-case convergence. Given a first-order optimization algorithm $\mathcal{A}$ we want to find the slowest convergence rate of $\mathcal{A}$ among all $f$ within a class of objective functions $\mathcal{C}$ and starting point $x_0 \in \mathbb{R}^d \setminus X_f^*$. That is equivalent to solving the following optimization problem at any step number $n$ :

$$\min_{f \in \mathcal{C}, (x_i)_{i \leq n} \in (\mathbb{R}^d)^{n+1}} \frac{\|x_0 - x_0^*\|_2}{\|x_n - x_n^*\|_2} \tag{5}$$
$$\text{s.t.} \quad \forall i \leq n-1, x_{i+1} = \mathcal{A}\left((x_0, f(x_0), \nabla f(x_0)), ..., (x_i, f(x_i), \nabla f(x_i))\right)$$

Directly searching for $f$ in the functional space is generally intractable, however if we can explicitly find the set $\mathcal{G}$ of all families $(x_i, f_i, g_i)_i$ such that $\exists f \in \mathcal{C}, \forall i, \nabla f(x_i) = g_i$ and $f(x_i) = f_i$ (we say that $f$ interpolates $(x_i, f_i, g_i)_i$), then problem (5) can be reduced to:

$$\min_{(x_i, g_i, f_i)_{i \leq n} \in (\mathbb{R}^d \times \mathbb{R}^d \times \mathbb{R})^{n+1}} \frac{\|x_0 - x_0^*\|_2}{\|x_n - x_n^*\|_2}$$
$$\text{s.t.} \quad \forall i \leq n-1, x_{i+1} = \mathcal{A}\left((x_0, f_0, g_0), ..., (x_i, f_i, g_i)\right) \tag{6}$$
$$\text{and} \quad (x_i, f_i, g_i)_{i \leq n} \in \mathcal{G}$$

In many cases (see Section 3), problem (6) is tractable and becomes a very powerful analysis tool providing lower bounds, upper bounds, and optimal tuning for different types of algorithms and assumptions used. A crucial and difficult component of this analysis is to formulate the *interpolation*

*conditions*, that is the necessary and sufficient conditions for a family $(x_i, f_i, g_i)_i$ to belong in $\mathcal{G}$. Driven by these motivations, we now establish the interpolation conditions for $\mathrm{RSI}^- \cap \mathrm{EB}^+$.

In Theorem 1, we introduce the necessary and sufficient conditions to interpolate a family $(x_i, g_i)$, without considering the function values $(f_i)_i$. This theorem could be used to find the worst case convergence rate over all first-order algorithms that ignore function values. However, in Corollary 1, we deduce from Theorem 1 sufficient (but not necessary) conditions to interpolate a family $(x_i, f_i, g_i)$. These conditions allow us in Section 5 to find a lower bound on the worst-case convergence rate for *all* first-order algorithms (including algorithms that have access to function values information), which we know is tight thanks to Prop 1.

---

**Theorem 1 (Interpolation conditions)** *Let* $(x_i, g_i)_{i \leq n} \in \left(\mathbb{R}^d \times \mathbb{R}^d\right)^{n+1}$, *such that the* $x_i$ *are separate points.*

*Then,* $\forall \mu, L > 0$:
$$\exists f \in \mathrm{RSI}^-(\mu) \cap \mathrm{EB}^+(L), \ s.t. \ \forall i, \ \nabla f(x_i) = g_i$$
$$\Updownarrow$$
$$\exists X^* \subseteq \mathbb{R}^d \ convex, \ s.t. \ \forall i,$$

$$\|g_i\|_2 \leq L \|x_i - x_i^*\|_2 \quad and \quad \langle g_i \mid x_i - x_i^* \rangle \geq \mu \|x_i - x_i^*\|_2^2, \tag{7}$$

*where* $x_i^*$ *is the orthogonal projection of* $x_i$ *onto* $X^*$.

---

*Proof.* In order to preserve concision and clarity, we will only present the broad outline of the proof here. For a complete technical proof, see Appendix B.

The direct implication is trivial as it is a direct application of $\mathrm{RSI}^-(\mu)$ and $\mathrm{EB}^+(L)$ definitions to $f$ in $(x_i)_i$. For the reverse implication, we will construct a function $f_{\epsilon,\beta}$ that interpolates $(x_i, g_i)_i$. The function $f_{\epsilon,\beta}$ is a quadratic everywhere except in the spheres of radius $\epsilon$ around each $x_i$, $\epsilon$ being small enough for these spheres to never intersect. Inside the sphere of radius $\epsilon$ around a given $x_i$, $f_{\epsilon,\beta}$ will be perturbed by adding a term $\lambda(\|x - x_i\|_2)h(x)$ where $h$ is affine in $x$, and $\lambda$ is a scaling term so that $\lambda(\epsilon) = 0$ at the border of the sphere, and $\lambda(0) = 1$ in its center $x_i$.

The key is to find a function $\lambda$ that preserves the properties of $\mathrm{RSI}^-(\mu)$ and $\mathrm{EB}^+(L)$. We use

$$\lambda_{\epsilon,\beta}(u) = \frac{1 + \cos\left(\pi \frac{u^\beta}{\epsilon^\beta}\right)}{2} \tag{8}$$

And our construction $f_{\epsilon,\beta}$ is given by:

$$f_{\epsilon,\beta}(x) = \begin{cases} \frac{\mu+L}{4} \|x - x^*\|_2^2 & \text{if } \forall i, \|x - x_i\|_2 \geq \epsilon \\ \frac{\mu+L}{4} \|x - x^*\|_2^2 + \lambda_{\epsilon,\beta}\left(\|x - x_i\|_2\right) \left\langle g_i - \frac{\mu+L}{2}(x_i - x_i^*) \mid x - x_i \right\rangle & \text{if } \exists i, \|x - x_i\|_2 < \epsilon \end{cases} \tag{9}$$

The rest of the proof is to use the Taylor expansions of $f_{\epsilon,\beta}$ to show that for sufficiently small $\epsilon$ and $\beta$, $f_{\epsilon,\beta}$ will belong in $\mathrm{RSI}^-(\mu)$ and $\mathrm{EB}^+(L)$ (see Appendix B).

∎

**Corollary 1** *Let $(x_i, f_i, g_i)_{i \leq n} \in (\mathbb{R}^d \times \mathbb{R} \times \mathbb{R}^d)^{n+1}$, such that the $x_i$ are separate points. Then, $\forall \mu, L > 0$:*

$$\exists X^* \subseteq \mathbb{R}^d \ convex, \ s.t. \ \forall i,$$

$$\|g_i\|_2 \leq L \|x_i - x_i^*\|_2$$
$$\langle g_i \mid x_i - x_i^* \rangle \geq \mu \|x_i - x_i^*\|_2^2 \tag{10}$$
$$f_i = \frac{\mu + L}{4} \|x_i - x_i^*\|_2^2$$

$$\Downarrow$$

$$\exists f \in \mathrm{RSI}^-(\mu) \cap \mathrm{EB}^+(L), \ s.t. \ \forall i, \ \nabla f(x_i) = g_i \quad and \quad f(x_i) = f_i,$$

*where $x_i^*$ is the orthogonal projection of $x_i$ onto $X^*$.*

*Proof.* We simply use the function $f_{\epsilon,\beta}$ from the proof of Theorem 1 and note that $\forall i, f_{\epsilon,\beta}(x_i) = \frac{\mu+L}{4} \|x_i - x_i^*\|_2^2$.

∎

Theorem 1 and Corollary 1 are the key elements to prove the optimality of gradient descent on $\mathrm{RSI}^-$ and $\mathrm{EB}^+$ among all first-order algorithms (see Section 5).

**Remark 4.1** *The interpolation conditions in Theorem 1 are independent, in the sense that a family $(x_i, g_i)_i$ admits an interpolation in $\mathrm{RSI}^-(\mu) \cap \mathrm{EB}^+(L)$ if and only if each $\{(x_i, g_i)\}$ admits an interpolation in $\mathrm{RSI}^-(\mu) \cap \mathrm{EB}^+(L)$. Similarly, the sufficient interpolation conditions in Corollary 1 are also independent.*

This property of independent interpolation conditions drastically simplifies convergence analysis and is the main reason we are able to analytically derive a lower bound in Section 5. Indeed, given an interpolable family $(x_i, f_i, g_i)_{i \leq n}$ for a given set $X^*$, it is sufficient to show that $(x_{n+1}, f_{n+1}, g_{n+1})$ is interpolable with $X^*$ to prove that the entire family $(x_i, f_i, g_i)_{i \leq n+1}$ can be interpolated. It is thus simple to find the set of interpolable $(f_{n+1}, g_{n+1})$ given $X^*$, $x_{n+1}$, and an interpolable $(x_i, f_i, g_i)_{i \leq n}$.

## 5 Lower bound on the convergence rate

In this Section we derive a lower bound on the convergence rate of first-order algorithms on $\mathrm{RSI}^-$ and $\mathrm{EB}^+$. This lower bounds applies under the assumption that the number of steps taken is smaller than the number of dimension $d$. This assumption is frequent in the literature (e.g. [5]) and not constraining for high-dimensional optimization. The observed gradients of the worst-case functions for optimal algorithms are typically orthogonal to one another (see [6]) which is not possible when the number of steps becomes larger than the dimension $d$. We conjecture that when not bounding the number of steps, it is possible to achieve an asymptotic rate in $O\left(2^{-\frac{n}{d}}\right)$ which would be better than the usual rates obtained for very ill-conditioned functions, while having little to no practical uses due to the bad convergence properties on a lower number of steps.

We now introduce Lemma 1, which is the cornerstone of the proof of Theorem 2:

**Lemma 1** *Let $\mu > 0$ and $L > \mu$. Let $\alpha_0 \in \left[ \frac{\mu}{L^2}, \max\left( \frac{\mu}{L^2}, \frac{1}{2\mu} \right) \right]$. For any first-order optimization algorithm $\mathcal{A}$ and starting point $x_0 \in \mathbb{R}^d$, there exists $(g_i)_{i \leq d-2} \in \mathbb{R}^d$, $(f_i)_{i \leq d-2} \in \mathbb{R}$ and $\mathcal{S}_{d-2} \subseteq \mathcal{S}_{d-1} \subseteq \cdots \subseteq \mathcal{S}_0 \subseteq \mathbb{R}^d$ such that:*

1. *$\forall i \leq d-2$, there exists a $(d-i-1)$-dimensional affine space $\mathcal{H}_i$ containing $\mathcal{S}_i$ and in which $\mathcal{S}_i$ is a $(d-i-2)$-sphere of radius $r_i = \sqrt{\frac{\alpha_0}{\mu} - \alpha_0^2} \|g_0\|_2 \left( 1 - \frac{\mu^2}{L^2} \right)^{\frac{i}{2}}$ and center $c_i \in \mathcal{H}_i$.*

2. *Let $(x_i)_i$ be the iterates generated by $\mathcal{A}$ starting from $x_0$ and reading gradients $(g_i)_i$ and function values $(f_i)_i$, then for any $i \leq d-2$ and any $x \in \mathcal{S}_i$, there exists a function $f$ in $\mathrm{RSI}^-(\mu) \cap \mathrm{EB}^+(L)$ minimized by $\{x\}$ that interpolates $(x_j, f_j, g_j)_{j \leq i}$.*

*Proof.* In order to preserve concision and clarity, we will only present the broad outline of the proof here. For a complete technical proof, see Appendix C.

We construct the sequence iteratively. For initialisation, we take any non-zero $g_0$, set $f_0 = \frac{\mu+L}{4\mu} \alpha_0 \|g_0\|_2^2$, $c_0 = x_0 - \alpha_0 g_0$, and finally

$$\mathcal{S}_0 = \left\{ x \in \mathbb{R}^d \,\middle|\, \langle x - c_0 \mid g_0 \rangle = 0 \right\} \cap \left\{ x \in \mathbb{R}^d \,\middle|\, \|x - c_0\|_2 = \sqrt{\frac{\alpha_0}{\mu} - \alpha_0^2} \|g_0\|_2 \right\}$$

Then assuming we have a sequence $(f_j, g_j, \mathcal{S}_j)_{j \leq i < (d-2)}$ respecting the conditions of the Lemma, noting $\mathcal{H}_i$ the $d-i-1$ dimensional affine space in which $\mathcal{S}_i$ is a $(d-i-2)$ dimensional sphere, and $x_{i+1}$ the $(i+1)$-th iterate returned by $\mathcal{A}$. Let $h_{i+1}$ the orthogonal projection of $x_{i+1}$ into $\mathcal{H}_i$.

If $h_{i+1} \neq c_i$, let $v = \frac{(h_{i+1} - c_i)}{\|h_{i+1} - c_i\|_2}$. If $h_{i+1} = c_i$, let $s \in \mathcal{S}_i$ and $v = \frac{(s - c_i)}{\|s - c_i\|_2}$.

We then construct:

$$c_{i+1} = c_i - \frac{\mu}{L} r_i v$$

$$f_{i+1} = \frac{\mu + L}{4} \left( \|x_{i+1} - c_{i+1}\|_2^2 + (1 - \frac{\mu^2}{L^2}) r_i^2 \right)$$

$$g_{i+1} = L \frac{\|x_{i+1} - x^*\|_2}{\|x_{i+1} - c_{i+1}\|_2} (x_{i+1} - c_{i+1})$$

$$\mathcal{H}_{i+1} = \left\{ x \in \mathcal{H}_i \mid \langle x - c_i \mid v \rangle = -\frac{\mu}{L} r_i \right\}$$

$$\mathcal{S}_{i+1} = \mathcal{S}_i \cap \mathcal{H}_{i+1}$$

We verify in Appendix C that this construction respects the properties of Lemma 1.

∎

We can now introduce Theorem 2 which gives us a lower bound on the worst-case convergence rate of any first-order algorithm on $\mathrm{RSI}^-(\mu)$ and $\mathrm{EB}^+(L)$.

**Theorem 2 (Lower bound on $\mathrm{RSI}^- \cap \mathrm{EB}^+$)** *Let $\mathcal{A}$ be any first-order algorithm on $\mathbb{R}^d$, $\mu > 0$ and $L \geq \mu$. For any $x_0 \in \mathbb{R}^d$, there exists $x^* \in \mathbb{R}^d$ and a function $f$ in $\mathrm{RSI}^-(\mu) \cap \mathrm{EB}^+(L)$ minimized by $X^* = \{x^*\}$ such that*

$$\forall i \leq d-1, \|x_i - x_i^*\|_2^2 \geq \left(1 - \frac{\mu^2}{L^2}\right)^i \|x_0 - x_0^*\|_2^2 \tag{11}$$

*Furthermore, if $\frac{L}{\mu} \geq \sqrt{2}$, there exists $h$ in $\mathrm{RSI}^-(\mu) \cap \mathrm{EB}^+(L)$ minimized by $X^* = \{x^*\}$ such that*

$$\forall i \leq d-1, \|z_i - z_i^*\|_2^2 \geq \frac{\|\nabla h(z_0)\|_2^2}{4\mu^2} \left(1 - \frac{\mu^2}{L^2}\right)^{i-1} \tag{12}$$

*where $(x_i)$ (resp. $(z_i)$) is the trajectory obtained by applying $\mathcal{A}$ to $f$ (resp. $h$) starting in $x_0$.*

Note that since $X^*$ is a singleton, $\forall i, x_i^* = z_i^* = x^*$.

*Proof.* If $L = \mu$, then the inequalities are trivial from the positivity of the norm. If $L > \mu$, let $(g_i, f_i, \mathcal{S}_i)_{i \leq (d-2)}$ be the sequence introduced in Lemma 1 for $\alpha_0 \in \left[\frac{\mu}{L^2}, \max\left(\frac{\mu}{L^2}, \frac{1}{2\mu}\right)\right]$. Let us note that for any $x \in \mathcal{S}_0$, $\|x_0 - x\|_2^2 = \frac{\alpha_0 \|g_0\|_2^2}{\mu}$ (see initialisation in Appendix C).

Let $i \in \{1, \ldots, d-1\}$. $\mathcal{S}_{i-1}$ has radius $r_{i-1}$, thus there exists $x^* \in \mathcal{S}_{i-1}$ such that $\|x_i - x^*\|_2 \geq r_{i-1}$, and thus :

$$\|x_i - x^*\|_2^2 \geq r_{i-1}^2 = \left(\frac{\alpha_0}{\mu} - \alpha_0^2\right) \|g_0\|_2^2 \left(1 - \frac{\mu^2}{L^2}\right)^{i-1} \tag{13}$$

When setting $\alpha_0 = \frac{\mu}{L^2}$ and observing that $x^* \in \mathcal{S}_{i-1} \subseteq \mathcal{S}_0$ and thus $\|x_0 - x^*\|_2^2 = \frac{\alpha_0 \|g_0\|_2^2}{\mu}$ in (13), we obtain (11). If $\frac{L}{\mu} \geq \sqrt{2}$, we set $\alpha_0 = \frac{1}{2\mu}$ and (13) immediately yields (12).

∎

**Remark 5.1** *Since the lower bounds established in Theorem 2 are exactly matched by the convergence guarantees of gradient descent (see Prop 1), these bounds are tight and gradient descent is exactly optimal on $\mathrm{RSI}^-(\mu) \cap \mathrm{EB}^+(L)$. This is a concrete example of the sensitivity of theoretical optimality to the choice of complexity classes.*

## 5.1 Discussion

The first bound presented in Theorem 2 gives the optimal solution when trying to solve

$$\min_{\mathcal{A}} \max_{f, x_0} \frac{\|x_n - x_n^*\|_2}{\|x_0 - x_0^*\|_2} \tag{14}$$

While the second bound gives the optimal solution when trying to solve

$$\min_{\mathcal{A}} \max_{f, x_0} \frac{\|x_n - x_n^*\|_2}{\|\nabla f(x_0)\|_2} \tag{15}$$

For general smooth and convex functions, (15) will not have a solution (for any $\mathcal{A}$, the quantity will not have a worst case upper bound), which is why (14) has historically been the focus of optimization literature. However in practice, when (15) admits a solution, as is the case for $\mathrm{RSI}^-(\mu) \cap \mathrm{EB}^+(L)$, it fits practical motivations better than (14) : when starting from $x_0$ and observing an initial gradient $g_0$, the solution to (15) is the one that will minimize $\|x_n - x_n^*\|_2$ in the worst case. In comparison, for a fixed $\|\nabla f(x_0)\|_2$, the solution to (14) will be faster when $\|x_0 - x_0^*\|_2$ is small, and slower when $\|x_0 - x_0^*\|_2$ is large, leading to a slower worst-case convergence.

When $\frac{L}{\mu} > \sqrt{2}$, using a tuning of $\alpha_0 = \frac{1}{2\mu}$ on the first step instead of $\frac{\mu}{L^2}$ leads to a worst-case convergence of $\|x_n - x_n^*\|_2$ better by a constant $c = \frac{L^2}{2(L^2 - \mu^2)}$. While such small constant factor is often considered not impactful, the number of steps required to make up for this constant factor

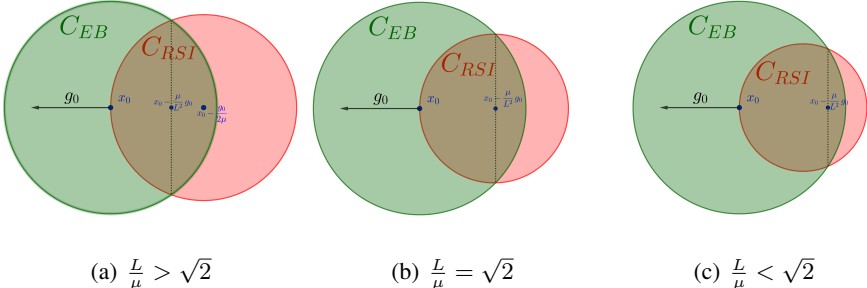

(a) $\frac{L}{\mu} > \sqrt{2}$       (b) $\frac{L}{\mu} = \sqrt{2}$       (c) $\frac{L}{\mu} < \sqrt{2}$

Figure 1: 2D representations of the possible positions of $x^*$ given $x_0$ and $g_0$, for three possible values of $\frac{L}{\mu}$. $x^*$ must be in $C_{RSI}$ (red) but not in $C_{EB}$ (green). When $\frac{L}{\mu} > \sqrt{2}$ (left), doing a larger step to reach the center of $C_{RSI}$ will minimize worst-case sub-optimality.

is $n = \frac{-\log(2)}{\log\left(1 - \frac{\mu^2}{L^2}\right)} - 1$, which yields $n \approx 68$ for $\frac{L}{\mu} = 10$ and $n \approx 6930$ for $\frac{L}{\mu} = 100$, and can thus become substantial on ill-conditioned functions.

Finally, we propose a geometric interpretation of the threshold $\frac{L}{\mu} = \sqrt{2}$. Given $x_0$ and $g_0$, $\mathrm{RSI}^-(\mu)$ requires $x^*$ to be within the circle of center $x_0 - \frac{g_0}{2\mu}$ and radius $\frac{\|g_0\|_2}{2\mu}$, while $\mathrm{EB}^+(L)$ requires $x^*$ to *not* be within the circle of center $x_0$ and radius $\frac{\|g_0\|_2}{L}$. In Figure 1 we show these circles for different values of $\frac{L}{\mu}$. When $\frac{L}{\mu} > \sqrt{2}$ (Figure 1(a)), $x_1 = x_0 - \frac{\mu}{L^2} g_0$ minimizes $\frac{\|x_1 - x^*\|_2}{\|x_0 - x^*\|_2}$ over all possible $x^*$, while $x_1 = x_0 - \frac{g_0}{2\mu}$ minimizes $\|x_1 - x^*\|_2$. As $\frac{L}{\mu}$ becomes smaller than $\sqrt{2}$ (Figure 1(b) and 1(c)), the same point $x_1 = x_0 - \frac{\mu}{L^2} g_0$ minimizes both quantities.

## 5.2 PEP experiment

Since we have found necessary and sufficient interpolation conditions in Theorem 1, we can use the PEP framework on $\mathrm{RSI}^- \cap \mathrm{EB}^+$ to confirm our results and derive the worst-case convergence rate of first-order algorithms. In Figure 2 we show the worst-case linear rate of convergence of Heavy Ball (HB) [29] on $\mathrm{RSI}^-(0.1) \cap \mathrm{EB}^+(1.0)$ for regularly sampled learning rate $\alpha$ and momentum $\beta$ (bright yellow means no linear convergence), generated with PEPit [9]. We remind the update rule of HB

$$x_{n+1} = x_n - \alpha \nabla f(x_n) + \beta(x_n - x_{n-1})$$

Since gradient descent is a special case of HB where $\beta = 0$, we observe as expected that the optimal rate of convergence is achieved for $\beta = 0$ and $\alpha = 0.1 = \frac{\mu}{L^2}$. Moreover, Figure 2 shows that momentum does not do well on $\mathrm{RSI}^- \cap \mathrm{EB}^+$ but gradient decent benefits from a relative robustness to the tuning of $\alpha$ : we get similar convergence rates for any $\alpha \in [0.05, 0.15]$.

## 6 Conclusion

Our main result is to prove that for any $\mu > 0$ and $L \geq \mu$, gradient descent is exactly optimal on the class of functions $\mathrm{RSI}^-(\mu) \cap \mathrm{EB}^+(L)$ (by exact optimality, we mean that the convergence guarantees of GD match the lower bound of worst-case performances exactly, without a constant factor of difference). This result confirms the observation in [12] that optimality is overly sensitive to the choice of assumptions, and should thus be considered with a lot of caution.

Interestingly, our analysis also identifies two similar notions of optimality, one of which suggests using a larger step size on the first iteration when the function is not particularly well-conditioned to improve worst-case convergence speed (see section 5.1).

We verified empirically that $RSI^- \cap EB^+$, with respect to the last iterate, are verified on the optimization paths of simple deep neural networks (c.f. Appendix A). This suggests that unlike usual alternatives which are known to not be verified on the highly non-convex loss landscapes of neural networks, convergence guarantees on $RSI^- \cap EB^+$ realistically apply to deep learning. On the

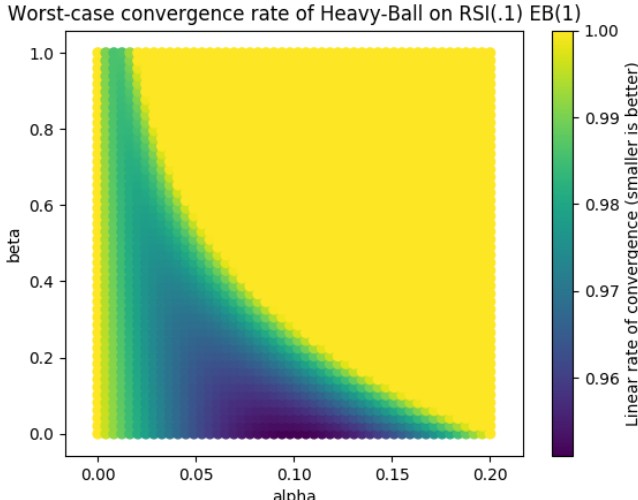

Figure 2: Worst-case linear convergence rate of heavy ball on $\mathrm{RSI}^-(0.1) \cap \mathrm{EB}^+(1)$ depending of its hyperparameters $\alpha$ and $\beta$, as calculated by PEP. The best rate is achieved for $\alpha = 0.1$ and $\beta = 0$.

other hand, the impossibility to accelerate the convergence rate on $RSI^- \cap EB^+$ implies that these assumptions are insufficient to explain the empirical successes of accelerated methods.

For the scope of this work we have focused on worst-case convergence analysis. While in practice average-case convergence rates are more insightful than their worst-case counterparts, such results are rare due to the necessity of defining a reasonable distribution on the considered class of functions, which is generally unfeasible. While such distribution on $\mathrm{RSI}^- \cap \mathrm{EB}^+$ is equally difficult to define, it should be feasible to define instead a reasonable distribution of the observed gradient for a given sampling point $x$ and nearest minima $x^*$, e.g. an uniform distribution over the (simple) set of possible gradients. Such approach is conceivable with $\mathrm{RSI}^- \cap \mathrm{EB}^+$ only because the interpolation conditions are independent (see remark 4.1), and thus we can easily make sure that any set of gradients sampled from this distribution can be interpolated within the class. While the distribution will necessarily be arbitrary, we believe such analysis could yield very useful insights and $\mathrm{RSI}^- \cap \mathrm{EB}^+$ is a rare opportunity to follow this approach.

Finally, many alternative conditions have been introduced in the literature (see Section 3), for which optimality results are still unknown. The PEP framework is a powerful tool to study conditions for which we can determine sufficient and necessary interpolation conditions, which would improve our understanding of first-order algorithm properties and tuning on a wide variety of objective function classes.

## Acknowledgments and Disclosure of Funding

The authors would like to thank Leonard Boussioux, for useful discussions and feedback. Ioannis Mitliagkas acknowledges support by an NSERC Discovery grant (RGPIN-2019-06512), a Samsung grant and a Canada CIFAR AI chair.

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
