# A   Optimization paths of neural networks are in $RSI^- \cap EB^+$

In this section we present a short experiment motivating $RSI^- \cap EB^+$ as useful assumptions for the study of optimization of neural networks. The goal is to estimate whether the gradients seen by training neural networks are interpolable by a function $f \in RSI^- \cap EB^+$. We propose the following process :

1. Set random seed S

2. Train a ResNet18 on CIFAR10 until convergence, and save the last iterate $x^*$

3. Reset random seed to S

4. Train a ResNet18 on CIFAR10, and at each iteration $i$, sample batch $B$ and measure $RSI_i = \frac{\langle \nabla f_B(x_i) | x_i - x^* \rangle}{\|x_i - x^*\|_2^2}$ and $EB_i = \frac{\|\nabla f_B(x_i)\|_2}{\|x_i - x^*\|_2}$

Where $\nabla f_B(x)$ is the gradient of the loss function on minibatch $B$, $x_i$ is the value of the weight at iteration $i$, and $x^*$ is the value of the last iterate measured at step 2.

Due to resetting the seed to a same value, the two training runs will be identical. We consider the last iterate $x^*$ to approximate a local minima, and $RSI_i$ and $EB_i$ will indicate whether the gradients seen during optimization are compatible with $RSI^- \cap EB^+$ with respect to that minima.

Even if the full-batch objective function that we intend to optimize is in $RSI^- \cap EB^+$, it is possible for $RSI_i$ to be negative due to the variance w.r.t the sampling of the minibatch $B$. However, we observe empirically that despite this, $RSI_i$ is lower bounded by a strictly positive value for every single iteration, without exception. This behavior is consistent across optimization algorithms (LARS, SGD without momentum, SGD with momentum, and ADAM) and initializations. For simplicity, we present here the results using SGD without momentum, with learning rate 0.1 and batch size $|B| = 1000$, and train for 360 epochs.

**Results:** we report the log of training loss in Figure 3 and the measured $RSI_i$ and $EB_i$ in Figure 4. Moreover, in order to better observe the behavior of $RSI_i$ and $EB_i$ outside of the initial peak, we report in Figure 5 the measured $RSI_i$ and $EB_i$ starting at epoch 30. We observe that $EB_i$ is upper bounded by $L = 2.303$ and $RSI_i$ is lower bounded by $\mu = 0.0010$, with a resulting condition number $\kappa = \frac{L}{\mu} = 2196.0$. Both have a significant peak at the beginning of training, justifying the popular use of learning rate warm-ups. When measuring bounds after epoch 30, we obtain $L = 0.2308$ resulting in a condition number $\kappa = 220.1$.

Surprisingly, despite the variance induced by minibatch sampling, the observed $RSI_i$ are all lower bounded by $\mu = 0.0010 > 0$. In particular, due to the necessary and sufficient conditions of $RSI^- \cap EB^+$ (See Section 4), it is guaranteed that there exists a function $f \in RSI^- \cap EB^+$ which exactly interpolates the gradients seen by the optimizer. And therefore, the convergence guarantees of $RSI^- \cap EB^+$ naturally apply to the optimization of neural networks in this setting.

Note that we do not claim that the objective function is in $RSI^- \cap EB^+$, which seems unlikely, but that the iterates explored by first-order algorithms are interpolable by functions in $RSI^- \cap EB^+$, including when sampling only part of the objective function through minibatches. This result strongly motivates the study of $RSI^- \cap EB^+$ as its guarantees apply to the optimization of neural network under assumptions empirically verified (at least in this simple setting).

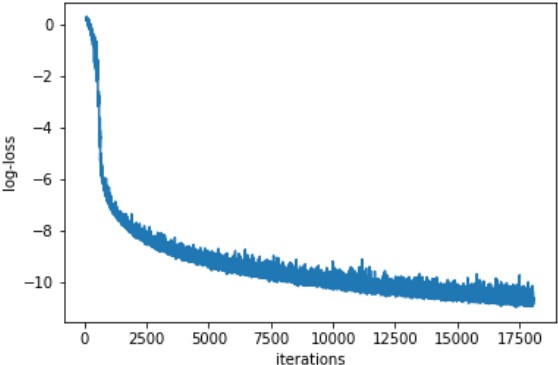

Figure 3: log-loss throughout training.

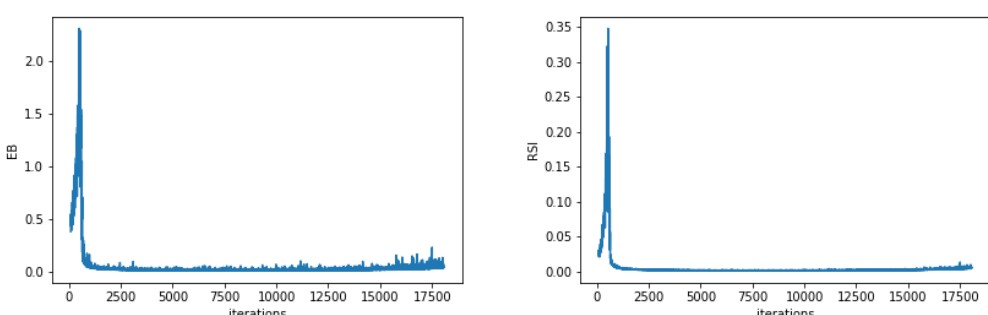

Figure 4: $RSI_i$ (right) and $EB_i$ (left) throughout training from epoch 0 to 360.

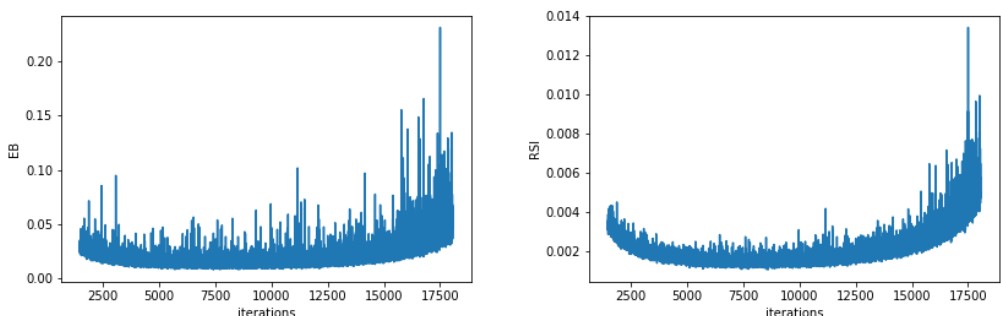

Figure 5: $RSI_i$ (right) and $EB_i$ (left) throughout training from epoch 30 to 360.

# B   Proof of Theorem 1

We start with two simple lemmas

**Lemma 2** *Let $X^*$ be a closed convex set, and $x^* \in X^*$ be the orthogonal projection of $x$ onto $X^*$. Then for any $y \in X^*$,*

$$\langle x^* - y \mid x - x^* \rangle \geq 0 \tag{16}$$

**Proof.** Let $y \in X^*$.

For $\theta \in [0, 1]$,
let $h(\theta) = \|x - ((1 - \theta)x^* + \theta y)\|_2^2 = \|x - x^*\|_2^2 + 2\theta \langle x - x^* \mid x^* - y \rangle + \theta^2 \|x^* - y\|_2^2$.

$h$ is differentiable and

$$h'(\theta) = 2 \langle x - x^* \mid x^* - y \rangle + 2\theta \|x^* - y\|_2^2 \tag{17}$$

Since $x^*$ is the orthogonal projection of $x$ onto $X^*$ and $\forall \theta \in [0, 1], (1 - \theta)x^* + \theta y \in X^*$, we have $\forall \theta \in [0, 1], h(\theta) \geq h(0)$, and thus $h'(0) \geq 0$. This concludes the proof of the Lemma thanks to (17).

∎

---

**Lemma 3** *If $x$ and $x_i$ are two points with respective orthogonal projections $x^*$ and $x_i^*$ on a closed convex set, then*

$$\|x - x^* - (x_i - x_i^*)\|_2 \leq 2 \|x - x_i\|_2 \tag{18}$$

---

**Proof.** As the case $x^* = x_i^*$ is trivial, we may assume that $x^* \neq x_i^*$.

Using lemma 2 twice, we get

$$0 \leq \langle x - x^* \mid x^* - x_i^* \rangle \tag{19}$$
$$0 \leq \langle x_i - x_i^* \mid x_i^* - x^* \rangle = \langle x_i^* - x_i \mid x^* - x_i^* \rangle \tag{20}$$

Adding the two inequalities, we get that

$$0 \leq \langle x - x^* - x_i + x_i^* \mid x^* - x_i^* \rangle = \langle x - x_i \mid x^* - x_i^* \rangle - \|x^* - x_i^*\|_2^2$$
$$\leq \|x - x_i\|_2 \|x^* - x_i^*\|_2 - \|x^* - x_i^*\|_2^2 \tag{21}$$

Since $x^* \neq x_i^*$, we obtain

$$\|x^* - x_i^*\|_2 \leq \|x - x_i\|_2 \tag{22}$$

And thus

$$\|x - x^* - (x_i - x_i^*)\|_2 \leq \|x - x_i\|_2 + \|x^* - x_i^*\|_2 \leq 2 \|x - x_i\|_2 \tag{23}$$

∎

---

We now move on to the proof of Theorem 1, that is :

Let $(x_i, g_i)_{i \leq n} \in \left(\mathbb{R}^d \times \mathbb{R}^d\right)^{n+1}$, such that the $x_i$ are separate points.

Then, $\forall \mu, L > 0$:
$$\exists f \in \mathrm{RSI}^-(\mu) \cap \mathrm{EB}^+(L), \ s.t. \ \forall i, \ \nabla f(x_i) = g_i$$

$$\Updownarrow$$

$$\exists X^* \subseteq \mathbb{R}^d \ convex, \ s.t. \ \forall i,$$

$$\|g_i\|_2 \leq L \|x_i - x_i^*\|_2 \quad \text{and} \quad \langle g_i \mid x_i - x_i^* \rangle \geq \mu \|x_i - x_i^*\|_2^2 \tag{24}$$

Where $x_i^*$ is the orthogonal projection of $x_i$ onto $X^*$.

**Proof.** The direct implication is trivial since the second property is simply the application of $\mathrm{RSI}^-$ and $\mathrm{EB}^+$ in each $x_i$. Let us now assume that (7) is verified.

First, let us note that if $L = \mu$, then we have $\forall i, g_i = \mu(x_i - x_i^*)$ and thus we can easily interpolate the $(x_i, g_i)$ using $f(x) = \frac{\mu}{2} \|x - x^*\|_2^2$. We now assume $L > \mu$.

If there is only one pair $(x_i, g_i)$, then we can simply use $f(x) = \langle g_i \mid x - x_i \rangle + \frac{\mu+L}{4} \|x - x_i\|_2^2$ to interpolate $(x_i, g_i)$. $f$ is then $\mu$-strongly convex and $L$-smooth so it is also in $RSI^-(\mu)$ and $EB^+(L)$. Let us now assume there are at least two pairs $(x_i, g_i)_i$. Let

$$\epsilon_0 = \frac{1}{2} \min_{i \neq j}(\|x_i - x_j\|_2) > 0 \tag{25}$$

By construction,

$$\forall x \in \mathbb{R}^d, (\exists i, \|x - x_i\|_2 < \epsilon_0) \Rightarrow \forall j \neq i, \|x - x_j\|_2 \geq \epsilon_0 \tag{26}$$

Moreover, if $\forall i, x_i \in X^*$, we can simply take $f(x) = \frac{\mu}{2}\|x - x^*\|_2^2$. Otherwise, let $\mathcal{I} = \{i \mid x_i \neq x_i^*\}$, and let $\epsilon_1 = \frac{1}{2}\min_{i \in \mathcal{I}}(\|x_i - x_i^*\|_2) > 0$

Let $\epsilon < \min(\epsilon_0, \epsilon_1)$ and $0 < \beta < \frac{1}{2}$. We introduce the function $\lambda_{\epsilon,\beta}$ from $[0, \epsilon]$ to $[0, 1]$ defined by :

$$\lambda_{\epsilon,\beta}(u) = \frac{1 + \cos\left(\pi\frac{u^\beta}{\epsilon^\beta}\right)}{2} \tag{27}$$

We finally introduce our interpolation function :

$$f_{\epsilon,\beta}(x) = \begin{cases} \frac{\mu+L}{4}\|x - x^*\|_2^2 & if \ \forall i, \|x - x_i\|_2 \geq \epsilon \\ \frac{\mu+L}{4}\|x - x^*\|_2^2 + \lambda_{\epsilon,\beta}(\|x - x_i\|_2)\left\langle g_i - \frac{\mu+L}{2}(x_i - x_i^*) \mid x - x_i \right\rangle & if \ \exists i, \|x - x_i\|_2 < \epsilon \end{cases} \tag{28}$$

First let us note that $f_{\epsilon,\beta}$ is properly defined : as stated in (26), there may be at most one $i$ such that $\|x - x_i\|_2 < \epsilon$. Moreover, $f_{\epsilon,\beta}$ is continuous because $\lambda_{\epsilon,\beta}(\epsilon) = 0$.

Since $\lambda_{\epsilon,\beta}(\epsilon) = 0$ and $\lambda'_{\epsilon,\beta}(\epsilon) = 0$, we can easily verify that for $x$ such that $\|x - x_i\|_2 = \epsilon$, $f_{\epsilon,\beta}$ is differentiable in $x$ with $\nabla f_{\epsilon,\beta}(x) = \frac{\mu+L}{2}(x - x^*)$. Thus $f_{\epsilon,\beta}$ is differentiable on $\mathbb{R}^d$. For any $x \in \mathbb{R}^d$ such that $\forall i, \|x - x_i\|_2 \geq \epsilon$, we have $\nabla f_{\epsilon,\beta}(x) = \frac{\mu+L}{2}(x - x^*)$ and thus trivially

$$\langle \nabla f_{\epsilon,\beta}(x) \mid x - x^* \rangle = \frac{\mu + L}{2}\|x - x^*\|_2^2 \geq \mu\|x - x^*\|_2^2 \tag{29}$$

$$\|\nabla f_{\epsilon,\beta}(x)\| = \frac{\mu + L}{2}\|x - x^*\|_2 \leq L\|x - x^*\|_2 \tag{30}$$

Let us now assume there is $i$ such that $\|x - x_i\|_2 < \epsilon$. If $x_i = x_i^*$, then $g_i = 0$ and $\nabla f(x) = \frac{\mu+L}{4}(x - x^*)$ and equations (29) and (30) are respected as well. Otherwise, we have $\|x - x^*\|_2 \geq \min_{i \in \mathcal{I}}(\|x_i - x_i^*\|_2) - \epsilon = \epsilon_1 - \epsilon > 0$
We then have, for $x \neq x_i$:

$$\begin{aligned}
\nabla f_{\epsilon,\beta}(x) &= \frac{\mu + L}{2}(x - x^*) + \lambda_{\epsilon,\beta}(\|x - x_i\|_2)\left(g_i - \frac{\mu + L}{2}(x_i - x_i^*)\right) \\
&\quad + \lambda'_{\epsilon,\beta}(\|x - x_i\|_2)\frac{x - x_i}{\|x - x_i\|_2}\left\langle g_i - \frac{\mu + L}{2}(x_i - x_i^*) \mid x - x_i \right\rangle \\
&= (1 - \lambda_{\epsilon,\beta}(\|x - x_i\|_2))\frac{\mu + L}{2}(x - x^*) + \lambda_{\epsilon,\beta}(\|x - x_i\|_2)g_i \tag{31} \\
&\quad + \lambda_{\epsilon,\beta}(\|x - x_i\|_2)\frac{\mu + L}{2}(x - x^* - (x_i - x_i^*)) \\
&\quad - \frac{\pi}{2}\frac{\|x - x_i\|_2^\beta}{\epsilon^\beta}\frac{\beta(x - x_i)}{\|x - x_i\|_2^2}sin(\pi\frac{\|x - x_i\|_2^\beta}{\epsilon^\beta})\left\langle g_i - \frac{\mu + L}{2}(x_i - x_i^*) \mid x - x_i \right\rangle
\end{aligned}$$

Since $X^*$ is a convex set (and closed by continuity of $f_{\epsilon,\beta}$), we have from Lemma 3 $\|(x - x^* - (x_i - x_i^*))\|_2 \leq 2\|x - x_i\|_2$.

To simplify notations, let us note $u = \|x - x_i\|_2$, $\lambda = \lambda_{\epsilon,\beta}(u)$, and

$r = -\frac{\pi}{2}\frac{u^\beta}{\epsilon^\beta}\frac{\beta(x-x_i)}{u^2}sin(\pi\frac{u^\beta}{\epsilon^\beta})\left\langle g_i - \frac{\mu+L}{2}(x_i - x_i^*) \mid x - x_i\right\rangle$.

We first want to upper bound $\|\nabla f_{\epsilon,\beta}\|_2$ using (31):

$$
\begin{aligned}
\|\nabla f_{\epsilon,\beta}(x)\|_2 &\le (1-\lambda)\frac{\mu+L}{2}\|x-x^*\|_2 + \lambda\|g_i\|_2 + (\mu+L)\lambda u + \|r\|_2 \\
&\le (1-\lambda)\frac{\mu+L}{2}\|x-x^*\|_2 + \lambda L(\|x-x^*\|_2 + \|x_i - x_i^*\| - \|x-x^*\|) + (\mu+L)\lambda u + \|r\|_2 \\
&\le L\|x-x^*\|_2 - (1-\lambda)\frac{L-\mu}{2}\|x-x^*\|_2 + (\mu+3L)\lambda u + \|r\|_2
\end{aligned}
$$
(32)

Moreover, the 3rd order remainder of the Taylor expansion of $cos(\pi\frac{u^\beta}{\epsilon^\beta})$ is $\frac{cos(c)}{4!}(\pi^4\frac{u^{4\beta}}{\epsilon^{4\beta}})$ for some $c$ in $[0, \pi\frac{u}{\epsilon}]$ and by upper bounding it we get $cos(\pi\frac{u^\beta}{\epsilon^\beta}) \le 1 - \frac{\pi^2 u^{2\beta}}{2\epsilon^{2\beta}} + \frac{\pi^4 u^{4\beta}}{24\epsilon^{4\beta}}$ and thus, for $\frac{u}{\epsilon} \le 1$,

$$
\begin{aligned}
-(1-\lambda)\frac{L-\mu}{2}\|x-x^*\|_2 &\le \left(-\frac{\pi^2 u^{2\beta}}{4\epsilon^{2\beta}} + \frac{\pi^4 u^{4\beta}}{48\epsilon^{4\beta}}\right)\frac{L-\mu}{2}\|x-x^*\|_2 \\
&\le -C_0\frac{u^{2\beta}}{\epsilon^{2\beta}}
\end{aligned}
$$
(33)

with $C_0 = \left(\frac{\pi^2}{4} - \frac{\pi^4}{48}\right)\frac{L-\mu}{2}\epsilon_1 > 0$

Furthermore, since $2\beta < 1$ and $\frac{u}{\epsilon} \le 1$, we can also bound

$$
(\mu+3L)\lambda u \le \epsilon(\mu+3L)\frac{u}{\epsilon} \le \epsilon C_1\frac{u^{2\beta}}{\epsilon^{2\beta}}
$$
(34)

with $C_1 = \mu + 3L > 0$

Finally, we can bound the last term using $sin(x) \le |x|$

$$
\begin{aligned}
\|r\|_2 &\le \beta\frac{\pi^2}{2}\frac{3L+\mu}{2}\|x_i - x_i^*\|_2\frac{u^{2\beta}}{\epsilon^{2\beta}} \\
&\le \beta C_2\frac{u^{2\beta}}{\epsilon^{2\beta}}
\end{aligned}
$$
(35)

with $C_2 = \frac{\pi^2}{2}\frac{3L+\mu}{2}\max_i(\|x_i - x_i^*\|_2)$

Finally, by choosing $\epsilon \le \frac{C_0}{2C_1}$ and $\beta \le \frac{C_0}{2C_2}$, and plugging (33), (34), (35) into (32), we get :

$$
\|\nabla f_{\epsilon,\beta}(x)\|_2 \le L\|x-x^*\|_2
$$

It only remains now to adequately lower bound $\langle\nabla f_{\epsilon,\beta}(x) \mid x - x^*\rangle$. We use the same method as before and keep the notations :

$$\langle \nabla f_{\epsilon,\beta}(x) \mid x - x^* \rangle = (1 - \lambda)\frac{\mu + L}{2} \|x - x^*\|_2^2 + \lambda \langle g_i \mid x - x^* \rangle$$

$$+ \lambda\frac{\mu + L}{2} \langle x - x^* - (x_i - x_i^*) \mid x - x^* \rangle + \langle r \mid x - x^* \rangle$$

$$\geq (1 - \lambda)\frac{\mu + L}{2} \|x - x^*\|_2^2 + \lambda\mu \|x_i - x_i^*\|_2^2 + \lambda \langle g_i \mid x - x^* - (x_i - x_i^*) \rangle$$

$$- \lambda(\mu + L)u \|x - x^*\|_2 - \|r\|_2 \|x - x^*\|_2$$

$$\geq (1 - \lambda)\frac{\mu + L}{2} \|x - x^*\|_2^2 - (1 - \lambda)\mu \|x - x^*\|_2^2 + \mu \|x - x^*\|_2^2$$

$$+ \lambda\mu \left( \|x_i - x_i^*\|_2^2 - \|x - x^*\|_2^2 \right) - 2u\lambda \|g_i\|_2$$

$$- \lambda(\mu + L)u \|x - x^*\|_2 - \|r\|_2 \|x - x^*\|_2$$

$$\geq \mu \|x - x^*\|_2^2 + C_0 \frac{u^{2\beta}}{\epsilon^{2\beta}} - 2\mu u(2 \|x_i - x_i^*\|_2 + u)$$

$$- 2\epsilon\lambda L \|x_i - x_i^*\|_2 \frac{u}{\epsilon} - \epsilon\lambda(\mu + L) \|x - x^*\|_2 \frac{u}{\epsilon} - \beta C_2 \|x - x^*\|_2 \frac{u^{2\beta}}{\epsilon^{2\beta}}$$

$$\geq \mu \|x - x^*\|_2^2 + (C_0 - \epsilon M_0 - \beta M_1)\frac{u^{2\beta}}{\epsilon^{2\beta}} \tag{36}$$

With $M_0 = 4(\mu + L)\max_i(\|x_i - x_i^*\|_2) + (L + 3\mu)\epsilon_1 > 0$
and $M_1 = C_2(\max_i(\|x_i - x_i^*\|_2) + \epsilon_1) > 0$

Therefore by taking $\epsilon \leq \frac{C_0}{2M_0}$ and $\beta \leq \frac{C_0}{2M_1}$, we guarantee

$$\langle \nabla f_{\epsilon,\beta}(x) \mid x - x^* \rangle \geq \mu \|x - x^*\|_2^2$$

Finally, for any $i$, we have :

$$\|x - x_i\|_2 \lambda'_{\epsilon,\beta}(\|x - x_i\|_2) = -\frac{\pi}{2} \frac{\|x - x_i\|_2^\beta}{\epsilon^\beta} \frac{\beta(x - x_i)}{\|x - x_i\|_2} \sin(\pi \frac{\|x - x_i\|_2^\beta}{\epsilon^\beta}) \tag{37}$$

Which goes to 0 as $x$ tends to $x_i$. Therefore using the definition of $f_{\epsilon,\beta}$ in (28) and the fact that $\left\langle g_i - \frac{\mu+L}{2}(x_i - x_i^*) \mid x - x_i \right\rangle$ is linear in $x - x_i$, we can conclude $\nabla f_{\epsilon,\beta}(x_i) = \frac{\mu+L}{2}(x_i - x_i^*) + \lambda_{\epsilon,\beta}(0)(g_i - \frac{\mu+L}{2}(x_i - x_i^*)) = g_i$.

We thus have proven that for sufficiently small $\epsilon$ and $\beta$, $\forall i, \nabla f_{\epsilon,\beta}(x_i) = g_i$, that for all $x$, $\langle \nabla f_{\epsilon,\beta}(x) \mid x - x^* \rangle \geq \mu \|x - x^*\|_2^2$ and $\|\nabla f_{\epsilon,\beta}(x)\|_2 \leq L \|x - x^*\|_2$. Therefore by definition, $f_{\epsilon,\beta}$ is in $\text{RSI}^-(\mu) \cap \text{EB}^+(L)$ and interpolates the $x_i, g_i$.

Which concludes the proof.

∎

## C  Proof of Lemma 1

Let $\mu > 0$ and $L > \mu$. Let $\alpha_0 \in \left[\frac{\mu}{L^2}, \max\left(\frac{\mu}{L^2}, \frac{1}{2\mu}\right)\right]$. For any first-order optimization algorithm $\mathcal{A}$ and starting point $x_0 \in \mathbb{R}^d$, there exists $(g_i)_{i \leq (d-2)} \in \mathbb{R}^d$, $(f_i)_{i \leq (d-2)} \in \mathbb{R}$ and $\mathcal{S}_{d-2} \subseteq \mathcal{S}_{d-1} \subseteq \cdots \subseteq \mathcal{S}_0 \subseteq \mathbb{R}^d$ such that:

1. $\forall i \leq d - 2$, there exists a $(d - i - 1)$-dimensional affine space $\mathcal{H}_i$ containing $\mathcal{S}_i$ and in which $\mathcal{S}_i$ is a $(d - i - 2)$-sphere of radius $r_i = \sqrt{\frac{\alpha_0}{\mu} - \alpha_0^2} \|g_0\|_2 \left(1 - \frac{\mu^2}{L^2}\right)^{\frac{i}{2}}$ and center $c_i \in \mathcal{H}_i$.

2. Let $(x_i)_i$ the iterates generated by $\mathcal{A}$ starting from $x_0$ and reading gradients $(g_i)_i$ and function values $(f_i)_i$, then for any $i \leq d - 2$ and any $x \in \mathcal{S}_i$, there exists a function $f$ in $\mathrm{RSI}^-(\mu) \cap \mathrm{EB}^+(L)$ minimized by $\{x\}$ that interpolates $(x_j, f_j, g_j)_{j \leq i}$.

**Proof.** For any first-order optimization algorithm $\mathcal{A}$ and starting point $x_0$, we are going to construct by induction the sequences $(g_i)_i$, $(f_i)_i$ and $(\mathcal{S}_i)_i$.

**Initialisation:** Let $g_0 \in \mathbb{R}^d \setminus \{0\}$. We can take any non-zero gradient as initialisation. Let $c_0 = x_0 - \alpha_0 g_0$, $f_0 = \frac{\mu+L}{4\mu} \alpha_0 \|g_0\|_2^2$, and

$$\mathcal{H}_0 = \{x \in \mathbb{R}^d \mid \langle x - c_0 \mid g_0 \rangle = 0\}$$

$\mathcal{H}_0$ is an hyperplane with dimension $d - 1$, and we finally introduce

$$\mathcal{S}_0 = \left\{ x \in \mathcal{H}_0 \mid \|x - c_0\|_2 = \sqrt{\frac{\alpha_0}{\mu} - \alpha_0^2} \|g_0\|_2 \right\}$$

By construction, $c_0 \in \mathcal{H}$ and $\mathcal{S}_0$ is the $(d-2)$-sphere in $\mathcal{H}_0$ of center $c_0$ and radius $r_0 = \sqrt{\frac{\alpha_0}{\mu} - \alpha_0^2} \|g_0\|_2$. Moreover, let $x^* \in \mathcal{S}_0$. We have

$$\|x^* - x_0\|_2^2 = \|x^* - c_0 + c_0 - x_0\|_2^2 = r_0^2 + \alpha_0^2 \|g_0\|_2^2 = \frac{\alpha_0 \|g_0\|_2^2}{\mu}$$

And thus $f_0 = \frac{\mu+L}{4} \|x^* - x_0\|_2^2$. We also have :

$$\langle g_0 \mid x_0 - x^* \rangle = \langle g_0 \mid x_0 - c_0 \rangle + \langle g_0 \mid c_0 - x^* \rangle = \alpha_0 \|g_0\|_2^2 + 0 = \mu \|x_0 - x^*\|_2^2$$

Finally, since $\alpha_0 \geq \frac{\mu}{L^2}$,

$$\|g_0\|_2^2 = \frac{\mu}{\alpha_0} \|x_0 - x^*\|_2^2 \leq L^2 \|x_0 - x^*\|_2^2$$

. Therefore all the sufficient conditions of Corollary 1 are verified, and there exists $f \in RSI^-(\mu) \cap EB^+(L)$ which is minimized by $\{x^*\}$ and interpolates $(x_0, f_0, g_0)$. This concludes the initialization.

**Induction:** Let us assume the existence of such $(f_j)_j$, $(g_j)_j$ and $(\mathcal{S}_j)_j$ up to step $i \leq d - 3$. Let $x_{i+1}$ be the iterate given by $\mathcal{A}$ after reading iterates $(x_j)_j$, function values $(f_j)_j$, and gradients $(g_j)_j$, let $\mathcal{H}_i$ the $(d-i-1)$-dimensional affine space in which $\mathcal{S}_i$ is a sphere, and let $c_i \in \mathcal{H}_i$ the center of the sphere $\mathcal{S}_i$.

If there exists $j \leq i$ such that $x_{i+1} = x_j$, then we simply return $g_{i+1} = g_j$ and $f_{i+1} = f_j$. We can take as $S_{i+1}$ any $(d-i-2)$-dimensional sphere of radius $r_{i+1}$ included in $S_i$, $\mathcal{H}_{i+1}$ its supporting affine space and $c_{i+1}$ its center. We now assume $\forall j \leq i, x_{i+1} \neq x_j$.

Let $h_{i+1}$ the orthogonal projection of $x_{i+1}$ into $\mathcal{H}_i$. If $h_{i+1} \neq c_i$, let $v = \frac{(h_{i+1} - c_i)}{\|h_{i+1} - c_i\|_2}$. If $h_{i+1} = c_i$, let $s \in \mathcal{S}_i$ and $v = \frac{(s - c_i)}{\|s - c_i\|_2}$.

Let

$$c_{i+1} = c_i - \frac{\mu}{L} r_i v$$

$$f_{i+1} = \frac{\mu + L}{4}\left(\|x_{i+1} - c_{i+1}\|_2^2 + (1 - \frac{\mu^2}{L^2})r_i^2\right)$$

$$g_{i+1} = L\frac{\|x_{i+1} - x^*\|_2}{\|x_{i+1} - c_{i+1}\|_2}(x_{i+1} - c_{i+1})$$

$$\mathcal{H}_{i+1} = \left\{x \in \mathcal{H}_i \mid \langle x - c_i \mid v \rangle = -\frac{\mu}{L} r_i\right\}$$

$$\mathcal{S}_{i+1} = \mathcal{S}_i \cap \mathcal{H}_{i+1}$$

$v$ is the difference between two points of $\mathcal{H}_i$, therefore it is one of the direction of $\mathcal{H}_i$, and since $c_i \in \mathcal{H}_i$, $\mathcal{H}_{i+1}$ indeed defines an affine subspace of $\mathcal{H}_i$ of dimension $(d - i - 2)$. Let $\mathcal{C}$ the sphere in $\mathcal{H}_{i+1}$ of center $c_{i+1} \in \mathcal{H}_{i+1}$ and radius $r_{i+1} = \sqrt{1 - \frac{\mu^2}{L^2}} r_i$. We now want to prove that $\mathcal{C} = \mathcal{S}_{i+1}$.

First, let $x \in \mathcal{H}_{i+1}$. Then

$$\begin{aligned}
\langle x - c_{i+1} \mid v \rangle &= \left\langle x - c_i + \frac{\mu}{L} r_i v \mid v \right\rangle \\
&= \langle x - c_i \mid v \rangle + \frac{\mu}{L} r_i \\
&= -\frac{\mu}{L} r_i + \frac{\mu}{L} r_i = 0
\end{aligned} \tag{38}$$

**i)** First, we show that $\mathcal{C} \subseteq \mathcal{S}_{i+1}$. Let $x \in \mathcal{C}$

$$\begin{aligned}
\|x - c_i\|_2^2 &= \left\| x - c_{i+1} - \frac{\mu}{L} r_i v \right\|_2^2 \\
&= \|x - c_{i+1}\|_2^2 + \frac{\mu^2}{L^2} r_i^2 - 2\frac{\mu}{L} r_i \langle x - c_{i+1} \mid v \rangle \\
&= \|x - c_{i+1}\|_2^2 + \frac{\mu^2}{L^2} r_i^2 && \text{using (38) since } x \in \mathcal{C} \subseteq \mathcal{H}_{i+1} \\
&= (1 - \frac{\mu^2}{L^2}) r_i^2 + \frac{\mu^2}{L^2} r_i^2 \\
&= r_i^2
\end{aligned} \tag{39}$$

since $x \in \mathcal{H}_{i+1} \subseteq \mathcal{H}_i$ and $\|x - c_i\|_2 = r_i$, $x \in \mathcal{S}_i$ and therefore $x \in \mathcal{S}_{i+1}$

**ii)** Conversely, we show that $\mathcal{S}_{i+1} \subseteq \mathcal{C}$. Let $x \in \mathcal{S}_{i+1}$.

$$\begin{aligned}
r_i^2 &= \|x - c_i\|_2^2 && \text{as } x \in \mathcal{S}_{i+1} \subseteq \mathcal{S}_i \\
&= \left\| x - c_{i+1} - \frac{\mu}{L} r_i v \right\|_2^2 \\
&= \|x - c_{i+1}\|_2^2 + \frac{\mu^2}{L^2} r_i^2 - 2\frac{\mu}{L} r_i \langle x - c_{i+1} \mid v \rangle \\
&= \|x - c_{i+1}\|_2^2 + \frac{\mu^2}{L^2} r_i^2 && \text{using (38) since } x \in \mathcal{S}_{i+1} \subseteq \mathcal{H}_{i+1}
\end{aligned} \tag{40}$$

from which we obtain that

$$\|x - c_{i+1}\|_2^2 = r_{i+1}^2 \tag{41}$$

So $x \in \mathcal{H}_{i+1}$ and $\|x - c_{i+1}\|_2 = r_{i+1}$, thus $x \in \mathcal{C}$. We have thus proved that $\mathcal{S}_{i+1}$ is indeed a $(d - i - 3)$-sphere in a $(d - i - 2)$ affine space with the desired radius and center which concludes the first item of the induction.

**We now want to prove the second item**. For $x^* \in \mathcal{S}_{i+1}$, $x_{i+1} - h_{i+1}$ is orthogonal to $\mathcal{H}_{i+1}$ due to being orthogonal to $\mathcal{H}_i$ by construction. $h_{i+1} - c_{i+1}$ is aligned with $v$ and thus is orthogonal to $\mathcal{H}_{i+1}$. Therefore, their sum $x_{i+1} - c_{i+1}$ is orthogonal to $\mathcal{H}_{i+1}$ and we get

$$\begin{aligned}
\|x_{i+1} - x^*\|_2^2 &= \|x_{i+1} - c_{i+1}\|_2^2 + \|c_{i+1} - x^*\|_2^2 \\
&= \|x_{i+1} - c_{i+1}\|_2^2 + r_{i+1}^2
\end{aligned} \tag{42}$$

And thus

$$f_{i+1} = \frac{\mu + L}{4} \|x_{i+1} - x^*\|_2^2 \tag{43}$$

Let $x^* \in \mathcal{S}_{i+1}$. Since $\mathcal{S}_{i+1} \subseteq \mathcal{S}_i$, then by recurrence hypothesis there exists an interpolation of the $(x_j, f_j, g_j)$ in $RSI^-(\mu) \cap EB^+(L)$ minimized by $x^*$, hence from Theorem 1,

$$\forall j \le i, \|g_j\|_2 \le L \|x_j - x^*\|_2 \quad \text{and} \quad \langle g_j \mid x_j - x^* \rangle \ge \mu \|x_j - x^*\|_2^2 \tag{44}$$

Moreover, by construction of $g_{i+1}$,

$$\|g_{i+1}\|_2 = L \|x_{i+1} - x^*\|_2 \tag{45}$$

Since $x_{i+1} - c_{i+1}$ is orthogonal to $\mathcal{H}_{i+1}$ and thus to $c_{i+1} - x^*$, we have

$$\langle x_{i+1} - c_{i+1} \mid x_{i+1} - x^* \rangle = \|c_{i+1} - x^*\|_2^2 \tag{46}$$

Besides, $x_{i+1} - c_{i+1}$ is orthogonal to $c_{i+1} - x^*$ and thus

$$\|x_{i+1} - x^*\|_2^2 = \|x_{i+1} - c_{i+1} + c_{i+1} - x^*\|_2^2 = \|x_{i+1} - c_{i+1}\|_2^2 + r_{i+1}^2 \tag{47}$$

By construction,

$$
\begin{aligned}
\|c_{i+1} - x_{i+1}\|_2 &\geq \|c_{i+1} - h_{i+1}\|_2 \\
&= \|c_{i+1} - c_i + c_i - h_{i+1}\|_2 \\
&= \left\| -\frac{\mu}{L} r_i v - \|h_{i+1} - c_i\|_2 \, v \right\|_2 \\
&= \frac{\mu}{L} r_i + \|h_{i+1} - c_i\|_2 \\
&\geq \frac{\mu}{L} r_i
\end{aligned}
\tag{48}
$$

and finally :

$$
\begin{aligned}
\frac{\langle g_{i+1} \mid x_{i+1} - x^* \rangle^2}{\mu^2 \|x_{i+1} - x^*\|_2^4} &= \frac{L^2}{\mu^2} \frac{\langle x_{i+1} - c_{i+1} \mid x_{i+1} - x^* \rangle^2}{\|x_{i+1} - x^*\|_2^2 \|x_{i+1} - c_{i+1}\|_2^2} \\
&= \frac{L^2}{\mu^2} \frac{\|x_{i+1} - c_{i+1}\|_2^2}{\|x_{i+1} - x^*\|_2^2} \qquad \text{using (46)} \\
&= \frac{L^2}{\mu^2} \frac{\|x_{i+1} - c_{i+1}\|_2^2}{\|x_{i+1} - c_{i+1}\|_2^2 + r_{i+1}^2} \qquad \text{using (47)} \\
&\geq \frac{L^2}{\mu^2} \frac{\frac{\mu^2}{L^2} r_i^2}{\frac{\mu^2}{L^2} r_i^2 + (1 - \frac{\mu^2}{L^2}) r_i^2} \qquad \text{using (48)} \\
&= 1
\end{aligned}
\tag{49}
$$

And thus

$$\langle g_{i+1} \mid x_{i+1} - x^* \rangle \geq \mu \|x_{i+1} - x^*\|_2^2 \tag{50}$$

Since $\forall j \leq i, x^* \in \mathcal{S}_j$, we also have

$$f_j = \frac{\mu + L}{4} \|x_j - x^*\|_2^2 \tag{51}$$

We finally apply Corollary 1 to all *unique* triples $(x_j, f_j, g_j)_{j \leq i+1}$ (which ensures by construction that all $x_j$ are distincts) which allows us to conclude from (43), (45), (44), (50) and (51) that there exists an interpolation in $RSI^-(\mu) \cap EB^+(L)$ that is minimized by $\{x^*\}$, proving the second item of the induction and thus concluding the proof.

∎