# OpenReview forum: "Gradient Descent Is Optimal Under Lower Restricted Secant Inequality And Upper Error Bound"
_NeurIPS.cc/2022/Conference — NeurIPS 2022 Accept_

### Official Review · Reviewer_gSLD · 2022-06-26

**Rating:** 5
**Confidence:** 4
**Soundness:** 4 excellent
**Presentation:** 4 excellent
**Contribution:** 2 fair

**Summary:**

In this work, the authors considered applying the gradient descent algorithm to functions in the intersection of RSI- and EB+. The intersection set is a generalization of the set of strongly convex and differentiable functions. A matching lower/upper bounds on the convergence rate is derived.

**Questions:**

Major:
(1) Line 78: since the stochastic gradient descent (SGD) method is considered for the optimization of neural networks, I wonder if the probability of escaping the attraction basin of local minima is sufficiently small. If the escaping probability is non-neglectable, the linear convergence may fail with certain probability and it is unsuitable to claim that RSI- and EB+ provide a guarantee on the linear convergence of SGD.

(2) In Thm.1 and Thm.2, the optimality gap is characterized by x_i - x_i^* and x_i - x^*. It would be better if the authors can provide a comparison between these two metrics, which is important to the tightness of lower/upper bounds.

(3) In the proof of Thm.2, it would be better to explicitly show why {x_i} are different points.

(3) Maybe I missed something, but I think functions in the intersection of RSI- and EB+ also satisfy the Polyak-Łojasiewicz (PL) condition. Under the PL condition, it is already known that descent algorithms with a (small) constant "step size" are optimal and converge linearly. Hence, the advantage of the bounds in this work lie in a matching lower/upper bound without a constant factor of difference, which seems marginal. I would suggest the authors provide a more detailed comparison with existing results on PL functions.

(4) In this experiments, the authors showed that the gradient descent method is better than the heavy-ball method. I wonder if this phenomenon is also observed on other functions in RSI- and EB+, or only appears on this special example. It would be better if more details of this experiments can be provided (e.g., which objective function is constructed).


Minor:
(1) Ln.20: it would be better to mention that alpha is the step size.

(2) Remark 2.2: I wonder if the authors can provide an example of non-convex functions that satisfy RSI-.

(3) Remark 2.4: the discussion after the first sentence seems unnecessary to me. When comparing RSI- and EB+ to other conditions, people usually assume that the same parameters (mu and L) are used. But I am okay if the authors want to keep this discussion.

(4) Ln.86: I think the Kurdyka-Łojasiewicz and the Polyak-Łojasiewicz conditions do not "take the form of a lower and an upper bound on properties of the objective function", since the gradient and the optimality gap appear on both sides of the inequality. If I am correct, it would be better to revise the statement on Ln.86.

(5) Ln.224: should "learning rates" be "convergence rates"?

**Limitations:**

See my comments in the "questions" section.

**Strengths And Weaknesses:**

In this work, the authors extended the results of classical convex optimization and showed that the gradient descent algorithm is optimal for functions in RSI- and EB+. I think the results are interesting to audiences in optimization and machine learning. Furthermore, the presentation of this paper is clear and easy to follow.

My major concern is on the importance of the contribution in this work. As the authors mentioned, there are lots of existing literature studying lower/upper conditions. Many of those conditions can also guarantee the linear convergence of gradient-based algorithms; also see my comments in the "questions" section. Hence, I am not sure if the contributions in this work are significant and would suggest the authors provide more discussions on the importance of the results.

---

> ### Author Response · Authors · 2022-08-02
> **Rebuttal**
>
> We thank the reviewer for their feedback. We address their concerns in the following:
>
> **Weakness:** While many works have derived linear convergence on all kinds of lower/upper condition pairs, it is extremely rare for such works to also provide lower bounds. A major contribution of our work is to show that gradient descent is exactly optimal among all first-order algorithms. For instance, it implies that acceleration can not be guaranteed without stronger assumptions. Moreover, it supports claims in Guille-Escuret et al. ([12] in the manuscript) that optimality results in first-order optimization must be taken with caution due to their sensibility to the choice of assumptions, and that Nesterov Accelerated Gradient may not always be the best choice even on strongly-convex and smooth problems. Finally, $RSI^-$ and $EB^+$ is a particularly wide class of functions in comparison to similar works, including even non-convex functions.
>
> **Major 1:** $RSI^-$ and $EB^+$ equations being verified on the sampled gradients wrt. the last iterate formally guarantees linear convergence to the last iterate. If the iterates were to fall into a local minima and eventually escape, it would indeed break linear convergence, but would also break $RSI^-$ and $EB^+$ equations (wrt. last iterate). In our experiments, we observe that this is not the case. While we expect that $RSI^-$ and $EB^+$ equations may not always be verified on more complex problems, we sustain our claim that in the case where they are verified (such as our experimental setting), it is sufficient to guarantee linear convergence. We believe this strongly supports the relevance of $RSI^-$ and $EB^+$ at least for simple applications of deep learning to image classification.
>
> **Major 2:** In Thm.2, we construct a function with a single global minima $X^*$={$x^*$}. Therefore, all projection onto $X^*$ are automatically equal to $x^*$ and $x_i^* = x^*$. The two formulations are thus perfectly equivalent and the bounds are perfectly tight. We propose to replace $x^*$ by $x_i^*$ in the formulation of Thm.2 to improve clarity.
>
> **Major 3.a:** The {$x_i$} in Thm.2 need not be distinct. We assume the reviewer is referencing the necessity of $x_i$ being separate points to apply Thm.1. This is not problematic : if the trajectory of algorithm A in Thm.2 contains duplicates, we simply apply Thm.1 to the set of unique tuples (removing duplicates). We propose to add this precision to the proof of Thm.2.
>
> **Major 3.b:**  We kindly ask the reviewer to specify to which line of work they are referencing. Indeed, such results would be impossible with PL alone, and would typically rely on an upper condition, which we guess to be smoothness. Smoothness is not implied by $RSI^-$ and $EB^+$ and may result in arbitrarily worse conditioning. Moreover, previous works ([12] in the manuscript) showed that smoothness is not a continuous condition, leading to pathological behaviours.
>
> **Major 4:** The experiment in section 5.2 is obtained by applying PEP to $RSI^-(0.1)$ and $EB^+(1)$. Each point on the figure corresponds to a specific tuning (alpha, beta) of heavy-ball, for which PEP has been used to determine the worst-case convergence rate and corresponding worst-case objective function in $RSI^-(0.1)$ and $EB^+(1)$. Therefore, there is not a single objective function being studied in Figure 2. Instead, each point corresponds to a worst-case objective function for this specific tuning of heavy-ball. There are of course functions in $RSI^-$ and $EB^+$ on which heavy-ball will perform better than gradient descent : e.g. quadratics. However, Figure 2 shows numerically that the tuning of heavy-ball that achieves the best worst-case convergence on $RSI^-(0.1)$ and $EB^+(1)$ is to set the momentum to zero and step size to 0.1, which is the theoretically prescribed gradient descent tuning in this setting.
>
> **Minor 1:** We will make that change.
>
> **Minor 2:** Here is a simple 1D example : $f(x) = 10x^2$ if $|x| < 1$; $f(x) = x^2 + 9$ if $|x| \geq 1$
>
> **Minor 4:** We follow the notations from Guille-Escuret et al. ([12] in the manuscript) which indeed consider Kurdyka-Łojasiewicz and the Polyak-Łojasiewicz as lower conditions (or upper if reversing the inequality direction). This is fairly intuitive as Polyak-Łojasiewicz implies $QG^-$ (lower bounded by a quadratic) but does not provide any upper bound on how fast the function can increase.
>
> **Minor 5:** It is indeed a mistake which we will correct

---

> > ### Comment · Reviewer_gSLD · 2022-08-09
> > **Post-rebuttal**
> >
> > I would like to thank the authors for responding to my questions! I have read all reviews and responses, and I would be happy to increase my score.

---

### Official Review · Reviewer_nnuj · 2022-07-11

**Rating:** 7
**Confidence:** 4
**Soundness:** 3 good
**Presentation:** 3 good
**Contribution:** 3 good

**Summary:**

This paper is a theoretical paper that studies the performance guarantees for gradient descent where the objective satisfies lower restricted secant inequality (weaker than the strong-convexity condition) and an upper error bound (weaker than the smoothness condition) so that this includes some non-convex objectives. The paper also proves that the gradient descent is optimal on this class of functions among all first-order algorithms.

**Questions:**

(1) In addition to the questions I mentioned above, one interesting thing I find is that the author(s) show that the gradient descent is optimal on this class of functions among all first-order algorithms. For $\mu$-strongly convex and $L$-smooth objectives, it is known that Nesterov's accelerated gradient descent and heavy ball method can accelerate. So for your class of functions, the gradient descent is already optimal? What is the intuition behind this? Since your class of functions also include non-convex functions, does this indicate that when you have non-convex objectives, gradient descent might outperform Nesterov's accelerated gradient descent or other momentum-based methods?

(2) In equation (9), write ''if'' in the text environment.

(3) In Corollary 1, in the last line, write ''where'' instead of ''Where'' and add a '','' in the equation before the last sentence. Same can be said about Theorem 1.

(4) In the first line in Theorem 2, it should be ''Let $\mathcal{A}$ be''.

(5) In the second line in the proof of Theorem 2, write ''be the sequence introduced in Lemma 1''.

(6) In reference [17], nesterov should be Nesterov and please check the other references as well.

**Limitations:**

It does not seem to me that the author(s) stated the limitations and potential negative societal impacts of their work.

**Strengths And Weaknesses:**

The paper is well written and is a solid theoretical paper. It is remarkable that the authors can show rigorously that the gradient descent is optimal on the class of functions that satisfy a lower restricted secant inequality and among an upper error bound all first-order algorithms.

The weakness of the paper is that even though the paper has a nice theoretical contribution, its practical importance and relevance is less clear to me. The class of objective functions that the author(s) study includes some non-convex functions, but when you have gradient descent, depending on the initialization, the algorithm can easily get stuck at a bad local minimum. Then the optimality discussed in the paper might be restrictive. Also for example, in practice, how do you measure your $\mu$ and $L$? In addition, the optimal stepsize for this class of objective functions is $\alpha=\mu/L^{2}$, which is smaller than the standard choice $\alpha=2/(\mu+L)$ for the $\mu$-strongly convex and $L$-smooth objective. I am wondering what is the intuition behind this because $\mu$-strong convexity and $L$-smoothness can be viewed as a special case under your assumption. What is the intuition that you need to choose smaller stepsize in your setup?

---

> ### Author Response · Authors · 2022-08-02
> **Rebuttal**
>
> We thank the reviewer for their feedback. We address their concerns in the following:
>
> **Weakness:** While $RSI^-$ $EB^+$ does include non-convex functions, $RSI^-$ prevents local minima, which should answer the reviewer’s first concern. We will clarify this in the paper. Measuring mu and L requires knowing the set of global minima, which is of course unrealistic in practice. While in Appendix A, we estimate mu and L by considering the final iterate as unique minima, the goal of our framework is not to design adaptive methods but rather build a sound theoretical framework to understand and justify the performances of first-order Algorithm on machine learning problems. For this purpose, empirical measurement of mu and L is not necessary.
> Regarding the stepsizes, we insist that the mu and L of $RSI^-$ and $EB^+$ are not the same as the ones from strong convexity and smoothness, thus it is not possible to determine which step size is larger without further information. In the case where there is a large gap in condition numbers (for a given function), the step size for $RSI^-$ $EB^+$ might be a lot larger than that of strong convexity and smoothness. On the other hand, if the condition numbers are close, then strong convexity and smoothness yield better regularity, allowing a larger step size than if we only had $RSI^-$ and $EB^+$.
>
> **Q1** Indeed, gradient descent is already optimal on $RSI^-$ and $EB^+$. This indicates that additional information is required to allow acceleration. Our intuition is that second-order regularity (e.g. strong-convexity, smoothness) is essential to achieve acceleration. However, this acceleration will depend on second-order conditioning. If that conditioning is bad, it may not be worth it to accelerate and better to perform gradient descent with a higher step size (due to better conditioning for $RSI^-$ and $EB^+$). It is an open question whether the two can be combined by leveraging badly-conditioned second-order properties to accelerate well-conditioned first-order properties.
>
> **Q2-6:** We thank the reviewer for their further suggestions and propose to incorporate it in our manuscript.

---

> > ### Comment · Reviewer_nnuj · 2022-08-09
> > **response to rebuttal**
> >
> > I thank the author(s) for clarifications and feedbacks. Please incorporate the changes in the revised version of your paper. Thank you!

---

### Official Review · Reviewer_Z4HC · 2022-07-11

**Rating:** 7
**Confidence:** 3
**Soundness:** 3 good
**Presentation:** 3 good
**Contribution:** 3 good

**Summary:**

This paper studied the complexity of gradient descent (GD) in RSI- and EB+ settings, which is more general than the common smooth and strongly convex case. The paper provided linear convergence results of GD in this case, and further derived the interpolation conditions and showed the optimality results of GD under this case based on the PEP framework.

**Questions:**

Basically I only have some minor questions:

1. You mentioned the optimal set $X^*$ should be convex, which means that the set $X^*$ is connected, right? I have the question because in Appendix A you tried to empirically shows that NN enjoys RSI- & EB+ style trajectory. But the landscape of NN may be complicated, and possibly attains disjoint (or nonconvex) optimal set. So I am confused on the statement "and therefore, the convergence guarantees of RSI− & EB+ naturally apply to the optimization of neural networks in this setting."

2. The independence of interpolation conditions look interesting. Following A. B. Taylor's thesis [32], it seems that the interpolation conditions for both strongly-convex smooth and nonconvex smooth are not independent (am I correct?). As a (kind of) "intermediate" setting, the RSI- & EB+ settings will enjoy the independence, which is a little counterintuitive to me. Is there any possible illustration, e.g., analysis technique? It may be better if authors can provide some discussion on the analysis compared to existing literature.

Some minor points:

1. Sec 5.2, I may suggest that authors should present the formal definition of HB algorithm, here the $\beta$ should distinguish with that of $f_{\epsilon, \beta}$
2. Corllary 1, change $\wedge$ to $\cap$?
3. Appendix A, why do you use "extrapolate" here, any difference compared to "interpolate"?

**Limitations:**

Yes

**Strengths And Weaknesses:**

Strength:
1. The conciseness of the results, also the analysis covers both upper and lower bound, which is a complete story.
2. The paper is well organized and easy to read.
3. The independence of the interpolation conditions is interesting.

Weakness:
1. The assumptions may be still a little restrictive (to be a little picky)

---

> ### Author Response · Authors · 2022-08-02
> **Rebuttal**
>
> We thank the reviewer for their feedback. We address their concerns in the following:
>
> **Q1:** Indeed, we assume $X^*$ convex and thus connected, neither of which are realistic assumptions for neural networks. In Appendix A, we measure $RSI^-$ and $EB^+$ equations with respect to the last iterate of the run (since we do not have direct access to the set of minima). Thus we do not guarantee convergence to a global minima, nor that the loss landscape necessarily belongs in $RSI^-$ and $EB^+$. However, our experiments show that the optimization trajectories satisfy $RSI^-$ and $EB^+$ equations with respect to the local minima $x^*$ to which the network eventually converges. With our results, this observation is sufficient to rigorously guarantee linear convergence to $x^*$ (with a rate that can be computed from mu and L).
>
> **Q2:** The reviewer is absolutely right : interpolation conditions for strong convexity and smoothness are not independent. If n points have been sampled, these conditions will correspond to an inequation for each pair of points, thus $O(n^2)$ equations, whereas $RSI^-$ and $EB^+$ only require a separate inequation per point, thus $O(n)$ equations. This independence of interpolation conditions is quite unique and extremely powerful for analytical purposes. One key reason it is possible is that neither $RSI$ or $EB$ constrain the function values $f$, only gradient values, which can have intense local deviations without impact outside a small neighbourhood. Some further intuition can be obtained from the proof of Theorem 1 (Appendix B).
>
> **minor points:** The use of the word extrapolate is accidental. We will replace it with interpolate, and will make the changes suggested by the reviewer.

---

> > ### Comment · Reviewer_Z4HC · 2022-08-09
> > **Post-rebuttal**
> >
> > Thank you for the response. I will keep my score.

---

### Official Review · Reviewer_tgSv · 2022-07-11

**Rating:** 7
**Confidence:** 3
**Soundness:** 3 good
**Presentation:** 2 fair
**Contribution:** 3 good

**Summary:**

Building on a series of work that explored alternative assumptions to smoothness and strong-convexity, that still lead to linear convergence of gradient descent but hold more generally, the submission presents an analysis of first order methods on a pair of such conditions; the restricted secant inequality lower bound and smoothness towards the optimum as the upper error bound. The results show that the worst-case convergence of gradient descent on the class of functions satisfying those assumptions is optimal, by providing a matching lower bound for first-order methods. The construction of the lower-bound deviates from the typical construction of quadratic functions using in smooth, convex optimization and uses that the restrictions imposed by the conditions are weaker than smoothness and strong-convexity to hide local deviations.

**Questions:**

The introduction relies heavily on the work of Guille-Escuret et al. ([12] in the manuscript) for the motivation of the investigation of the limits of the conditions studied. Motivations specific to the current submission are somewhat missing, and the conclusions are somewhat at odd with the motivations of Guille-Escuret et al., leaving me with some unaddressed questions. Those are not points against the results of the paper, and I believe it is a matter of editing to make the motivations clearer and can be addressed during a discussion period.

- Why focus on the specific conditions in this paper, among the many possible combinations of conditions, even if restricted only to the ones discussed by Guille-Escuret et al.? Have other stronger (in the sense that they would imply the studied condition, up to a change of constants) conditions already been studied?
- One of the motivations of Guille-Escuret et al. for the study of those conditions was to address the current limitations of the theory for accelerated methods, as in Polyak's Heavy Ball, where the parameter setting used for quadratics does not extend to smooth, strongly-convex functions. The result that Gradient Descent is optimal for this class of problem would suggest that the class considered is too large to be useful for the design of new methods, or contains pathological functions. For example, the construction given for the interpolation, while interesting, indicates that the function has arbitrary many local deviations that are unrelated to each other, leading to the question of whether this relaxation of smoothness/strong-convexity is a valid model to pursue?

**Applicability of the assumptions**

Some of the limitations of the work are in regard with the level of applicability of the results. The submission makes the point multiple time that, although the assumptions considered are implied by standard smoothness and strong-convexity assumptions, they are not strictly weaker as their condition number can improve.
Thus the improvement in the rates might be worth the degradation in the dependence on the new condition number, "provided we can obtain better constants under these conditions" (L65). This is reiterated in Remark 2.5. While I am sympathetic to this argument---for example, for least-squares, it is known that the strong-convexity constant can be 0 the Polyak-Lojasiewicz condition is always non-zero---the arguments of the paper would be strengthened by a simple, provable example of the gap between pairs of conditions on a problem of interest. Using the notation of Guille-Escuret et al. ([12]), on what problem do we see a benefit from using `(RSI-, EB+)` instead of smoothness/strong-convexity, that is not already captured by `(PL+, *SC-)`?

I acknowledge that the submission also discuss the applicability of the conditions to fitting neural networks (L80-82). However, I would recommend adding a qualifier after "an impressive feat given the highly non-convex nature of neural networks loss function" (L82) to avoid overstating their applicability, similar to the qualifying paragraph found in Appendix A after L420.


**Limitations:**

No concern beyond the lack of a clear-cut use-case for the applicability of the assumptions used here noted above.

**Strengths And Weaknesses:**

To my knowledge, the results presented are new and the proof technique used to obtain the lower bound is technically interesting. The overall presentation of the technical results are reasonably clear. My main concern is on the clarity of the motivations, which I expand on in the next section. Regarding significance, althought not explicitly connected to this line of research in the submission, the results are interesting for the study of the possibility of acceleration under looser conditions than smooth/strong-convex which have attracted attention for applications in ML and explaining the effectiveness of momentum-based methods. This study shows the limitations of some relaxations of smoothness and strong-convexity, showing that acceleration beyond gradient descent is not provable without additional structure.

---

> ### Author Response · Authors · 2022-08-02
> **Rebuttal**
>
> We thank the reviewer for their feedback. We address their concerns in the following:
>
> **Q1** A few other pairs of condition have been studied such as very recent work on Upper Quadratic Growth ($QG^+$) and convex [1], or to a larger extent $PL^-$ and smooth. However, most of them remain open problems. Perhaps the main reason to focus on $RSI^-$ and $EB^+$ is that many of the convergence rates reported in table 1 of Guille-Escuret et al. are inherited from the rate of $RSI^-$ and $EB^+$ (through implications between classes), hinting that this pair of conditions might hold particular meaning for first-order optimization. Another reason is that we were able to find necessary and sufficient interpolation conditions, which can be arduous. Moreover, they are independent, which makes a thorough theoretical analysis more accessible.
>
> [1] Goujaud, B. and Taylor, A. and Dieuleveut, A. Optimal first-order methods for convex functions with a quadratic upper bound. arXiv preprint arXiv:2205.15033. 2022 May 30.
>
> **Q2** This is an excellent remark. Our results suggest that second-order regularity is a key component to achieving acceleration. The empirical successes of momentum methods indicate that practical objectives probably have some level of second-order regularity, while Guille-Escuret et al. shows that building theory on second-order assumptions is unreliable, giving examples of pathological behaviors ignored by such theory. Perhaps momentum is efficient in practice, but not in the worst-case, suggesting it cannot be properly understood with the limitations of worst-case theoretical analysis. Perhaps the first-order properties and second-order properties can be simultaneously leveraged (with distinct constants) to prevent pathological behaviors while still allowing acceleration when possible.
> Moreover, the experiments in Appendix A suggest that the gradients sampled by SGD when optimizing a neural network might look a lot like the gradients deterministically measured from a function in $RSI^-$ and $EB^+$. The local deviations mentioned by the reviewer allow the sudden changes in gradient induced by the stochasticity of the sampling process. Therefore, $RSI^-$ and $EB^+$ might be particularly suited to capture the empirical properties of neural network loss landscapes.
>
>
> **Applicability:** Using the notation of Guille-Escuret et al. ([12]), on what problem do we see a benefit from using ($RSI^-$, $EB^+$) instead of smoothness/strong-convexity, that is not already captured by ($PL^+$, $*SC^-$)?:
>
> The challenge in providing such examples is that loss functions with a large gap in conditioning between $RSI^-$ $EB^+$ and e.g. strong-convexity and smoothness have non-trivial landscapes making the analytical derivation of their condition number intractable. This is why we instead conducted an experiment to empirically estimate the interpolability of neural net gradients by functions in $RSI^-$ $EB^+$, as motivation for this pair of conditions. We will follow the advice of the reviewer and rephrase the claim L82 to better reflect that our experiment does not show that neural net loss functions strictly belong in $RSI^-$ $EB^+$, but rather that the observed gradients are interpolable. This observation is sufficient however to rigorously apply convergence guarantees of $RSI^-$ $EB^+$ to the convergence of the neural net.

---

> > ### Comment · Reviewer_tgSv · 2022-08-09
> > **Post-rebuttal**
> >
> > Thank you for your reply.
> >
> > Post-rebuttal, the main common issue seems to be the implications of the contribution. I tend to agree that a weak point of the submission is that the consequences of the results are unclear, beyond that the convergence of GD is optimal on this class. But I think this can be addressed by a minor revision of a paragraph in the introduction and conclusion to explicitly address the impossibility result (Q2 above, details below) and I am raising my score.
> >
> > ---
> >
> > As other reviewers note, there are many works on alternative assumptions for the convergence rate of GD, to relax smoothness/strong-convexity. Beyond emphasizing the positive result that GD is optimal, the equivalent negative result that acceleration is impossible is a strong result that would make the submission shared more broadly. One of the hopes of this literature is that such guarantees would capture the behavior of optimization for modern ML models. The results show that this specific relaxation is insufficient to explain the benefits of momentum-based methods, as it not only allows for non-convex behavior but also pathological behavior that restricts acceleration. I encourage the authors to explicitly address this point to make sure it isn’t missed by readers.

---

> > > ### Author Response · Authors · 2022-08-09
> > > **Additional comment**
> > >
> > > Thank you for your very pertinent reply.
> > >
> > > We fully agree with your suggestion regarding the characterization of our contributions. We will improve our introduction and conclusion to explicitly reflect the implications of our results on impossibility of acceleration.
> > >
> > > As a side note, outside the scope of our submission but which may be of interest to the reviewer, we suspect that allowing acceleration while using more realistic conditions that are robust to the pathological behaviors described in Guille-Escuret et al. may be impossible under the scope of worst-case analysis. Momentum-based methods may only be efficient with high probability in modern ML applications (with reasonable assumptions), which will require a significant change in the typical worst-case framework used to study first-order optimization.

---

### Meta-Review · Area_Chair_gL3K · 2022-08-21

**Recommendation:** Accept
**Confidence:** Certain

**Metareview:**

A solid theoretical paper with fine execution that establishes the optimality of the vanilla gradient descent method in a class of functions extending the well-studied class of smooth and strongly convex functions. Please make sure to take into account the insightful feedback given by the reviewers in the revised version.

**Award:**

No

---

### Decision · Program_Chairs · 2022-09-14

Accept